


# Intercomparing radar data assimilation systems for ICE-POP 2018 snowfall cases

Ki-Hong Min[1,2], Kao-Shen Chung[3], Ji-Won Lee[1,2], Cheng-Rong You[3] and Gyuwon Lee[1,2]

[1]Department of Atmospheric Sciences, Kyungpook National University, Daegu, 41566, Korea
[2]Center for Atmospheric Remote Sensing, Kyungpook National University, Daegu, South Korea
[3]Department of Atmospheric Sciences, National Central University, Jhung-li, 320, Taiwan

*Correspondence to*: Ki-Hong Min (kmin@knu.ac.kr)

**Abstract.** Gangwon-do (GWD) has complex terrain and surface characteristics due to its location to the East Sea and the Taebaek Mountain range. This coastal location and rugged terrain can amplify snowfall mechanisms, making it challenging to

accurately predict the amount and location. This study compares two methods for assimilating radar data and analyzed snowfall prediction results. The two methods compared are local ensemble transform Kalman filter (LETKF) and three-dimensional variational (3DVAR) data assimilation (DA). LETKF improved the water vapor amount and temperature using the covariance of the ensemble members, but 3DVAR improved the water vapor mixing ratio and temperature through an operator that assumed the atmosphere was saturated when reflectivity was above a certain threshold. In 2018, to understand the snowfall in

GWD region and support the Pyeongchang Winter Olympic and Paralympic Games, a long-term heavy snow observation campaign was conducted. The International Collaborative Experiments for the 2018 Pyeongchang Olympic Games Projects (ICE-POP 2018) data are used to study and verify the numerical experiments. From the initial field verification using ICE-POP observation data (radiosonde), wind in LETKF was more accurately simulated than 3DVAR, but it underestimated the water vapor amount and temperature in the lower troposphere due to a lack of a water vapor and temperature observation

operator. Snowfall in GWD was less simulated in LETKF, whereas snowfall of 10.0 mm or more was simulated in 3DVAR, resulting in an error of 2.62 mm lower than LETKF. The results signify that water vapor assimilation is important in radar DA and significantly impacts precipitation forecasts, regardless of the DA method used. Therefore, it is necessary to apply observation operators for water vapor and temperature in radar DA.

## 1 Introduction

In winter in the Korean Peninsula, natural disasters, such as heavy snowfall, primarily cause property and human damage. The Gangwon-do (GWD) region is characterized by a large amount of precipitation due to heavy snowfall and long snowfall days compared to other regions in South Korea. Mechanisms generating heavy snowfall systems in the GWD region can be divided into heavy snow caused by a low-pressure system and an airmass modification-type snowfall system. Airmass modification-type snowfall is characterized by cold and dry continental airmass moving over relatively warm and humid sea

surfaces, receiving heat and water vapor, forming a snowfall system at sea, and causing heavy snow to the ground at physical



boundaries, such as mountain ranges or coasts. GWD region is characterized by heavy snowfall and many snowfall days caused by the airmass modification (Kim et al., 2005; Kim et al., 2021; Lee, 1994; Lee and Park, 1996; Nam et al., 2014; Tsai et al., 2018).

Because GWD area is adjacent to the East Sea and the Taebaek Mountains, meteorological phenomena are amplified (Fig.
1). The area of the East Sea is approximately 106 km$^2$, and the average depth is 1,800 m, providing relatively cool ocean winds to GWD region in summer, and relatively warm ocean winds in winter, serving as heat storage and supplying water vapor to the atmosphere. The snow clouds developing in the East Sea cause snowfall in GWD area and the Taebaek Mountains, and the snow accumulated in the mountain area serves as a reservoir for water resources until spring and helps prevent forest fires occurring along with strong winds in the dry season in spring (Murakami et al., 2003; Nam et al., 2014). The Taebaek
Mountains, which significantly influence the meteorological phenomena in the GWD region, are the longest mountain range on the Korean Peninsula, stretching over 600 km from Hamgyong Province, North Korea, to Busan, South Korea. The average elevation is 800–1,000m, and it has a gentle slope to the west but a steep slope to the east, where it meets the East Sea.

In late winter between February and March, when the Siberian high pressure extends southeast, crossing northern China and the East Sea, snowfall frequently occurs in the GWD area (Nam et al., 2014). As the Siberian high pressure weakens during
the late winter, a low-pressure system passes south of the Korean Peninsula, creating a cool northeasterly to easterly wind over the mild East Sea (Lee et al., 2012; Park et al. 2009). Eventually, snow clouds develop due to convective instability in the East Sea, inducing heavy snow in the GWD area (Lee & Lee, 2003).

Due to preceding studies, the prediction accuracy of snowfall cases dominated by a distinct synoptic condition has been considerably improved. The GWD region has a stronger effect from the East Sea and the mountain range because the slope
from the top of the Taebaek Mountains to the East Sea is steep. Therefore, to improve the weather forecast in a region adjacent to the sea in a narrow area, such as the Korean Peninsula, comprising complex topography, analysis of and understanding mesoscale meteorological phenomena, dominated by topographical effects, are required. However, in Korea, observation and analysis are still lacking. Therefore, target-oriented integrated intensive observation for understanding mesoscale meteorological effects due to mountain ranges and the East Sea effect in a natural laboratory with complex topography, such
as GWD area, is essential.

Snowfall studies in the GWD area lacked intensive observation to comprehensively interpret the snowfall mechanisms. However in 2018, before and during the Pyeongchang Winter Olympics, through the International Collaborative Experiment for Pyeongchang Olympic and Paralympics (ICE-POP 2018), a field campaign was conducted in this complex topography region to study microphysics and characteristics of solid precipitation growth and snowfall mechanisms (Kim et al., 2021;
Gehring et al., 2020). Numerous international organizations and universities (29 agencies from 12 countries), such as Castilla-La Mancha University in Spain, Ecole Polytechnique Fédérale de Lausanne in Switzerland, Austrian Meteorological Agency, National Central University in Taiwan, Colorado State University, National Aeronautics and Space Agency, National Oceanic and Atmospheric Administration, and National Center for Atmospheric Research (NCAR) in the USA, observed solid precipitation in winter through joint research and collaboration. Using data based on the international joint program, a detailed



investigation was conducted on the complexity of multiscale phenomena, topographical effects, ocean–land-surface
heterogeneity, and constraints of input data and temporal/spatial resolution, the challenges of quantitative forecasting
highlighted as limitations of winter snowfall prediction (Gehring et al., 2020, Jung et al., 2020; Lim et al., 2020; Kim et al.,
2021; Tsai et al., 2018).

Because radar data only include information on hydrometeors in the atmosphere and exclude information on water vapor,
several radar data assimilation (RDA) studies do not assimilate the water vapor mixing ratio (Chen et al., 2021; Liu et al., 2019;
Tong et al., 2020). Wang et al. (2013) developed a method to assimilate the saturated water vapor mixing ratio, assuming that
the area is saturated when the reflectivity above a certain threshold value above the cloud base is observed. Lee et al. (2022)
used cloud microphysical process analysis and revealed that water vapor significantly contributes to precipitation formation
compared to hydrometeors. Until now, research on new ways to improve the water vapor mixing ratio using radar data
continues (He et al., 2021; Pan et al., 2020).

Many RDA studies have utilized the three-dimensional variational (3DVAR) method to assimilate radar data (Ide et al. 1999;
Sugimoto et al., 2009; Maiello et al., 2014; Lee et al., 2020; Vendrasco et al., 2016; Wang et al., 2013). The optimal analysis
is obtained by minimizing the cost function through repetitive operations. This method is suitable for assimilating significant
amounts of radar data because it requires fewer computer resources and equation constraints than other data assimilation (DA)
methods. The limitations of 3DVAR however is that the model background error (BE) only considers the climatic BE
covariance and it was developed for large-scale observational data (Hamil and Snyder, 2002). That is, simulating the
continuously varying weather phenomena can be limited due to of the isotropic and homogeneous effect of 3DVAR method.
The local ensemble transform Kalman filter (LETKF) based on the Bayesian approach can provide more optimal analysis,
because LETKF generates many ensemble members (10-100) and calculates the current state BE considering errors associated
with the relationship of variables and between grid points using the ensemble members. Thus, the DA effect corresponds to
the flow of the atmosphere and model errors changing with time is considered (Stensrud and Gao, 2010). But this method has
the disadvantage of requiring more calculations than 3DVAR because it has to generate multiple analyses and prediction of
ensemble members. Further, sampling error can occur when the ensemble members are small. With the increase in computer
performance and resources, it is becoming more common in RDA research (Seko et al., 2011; Tsai et al., 2014; Wu et al., 2020;
Yang et al., 2022; You et al., 2020).

This study evaluates the improvement of precipitation forecasts using radar and in-situ observation data collected from ICE-
POP 2018 and compares 3DVAR and LETKF DA methods in simulating winter storm cases. The statistical verifications of
experimental results are calculated based on intensive observations during this period. Section 2 describes the data, DA
methods, experimental design, and model configurations. Section 3 analyzes increments in the initial field and hydrometeor
mixing ratios, radar reflectivity contoured frequency by altitude diagram (CFAD), and the verification statistics of forecasts
according to two DA methods. Section 4 summarizes the study and concludes based on the study results.



## 2 Data and Experiments

### 2.1 Observation data

Figure 2 shows the coverage map of radar sites and the location of observation data. In this study, S-band radar data from the
Ministry of Environment (MOE) and the Korea Meteorological Administration (KMA) are used. The site locations are
Baengnyeongdo, Gari Mountain, Gudeok Mountain, Go Mountain, Gwanak Mountain, Gwangdeok Mountain, Oseong
Mountain, Seong Mountain and Sobaek Mountain. The fuzzy-logic algorithm developed at Kyungpook National University
(KNU) was used to improve the quality of radar signal and remove anomalous electromagnetic waves and non-meteorological
echoes in the air (Ye et al., 2015). A composite field of constant altitude plan position indicator (CAPPI) was created by
interpolating the CAPPI data on a 200 m vertical and 3 km horizontal grid equal to the model grids. The 3D gridded radar field
was used as the DA input data. To evaluate the accuracy of the analysis, KMA and ICE-POP radiosondes were used. The
precipitation forecast accuracy was evaluated by comparing the model experiments and the KMA's automated weather stations
(AWSs) precipitation data.

### 2.2 Methods of DA and experiment design

#### 2.2.1 3DVAR DA method

The weather research and forecasting (WRF) model and its variational DA system is used assimilate radar data (Barker et al.,
2004; Barker et al., 2012). The 3DVAR system is conducted using iterative computations to find the minimum cost function
$J$ with a gradient descent method (Eq. 1).

$$J = J_o + J_b = \frac{1}{2}(\boldsymbol{y} - \boldsymbol{y_0})^T \boldsymbol{R^{-1}}(\boldsymbol{y} - \boldsymbol{y_0}) + \frac{1}{2}(\boldsymbol{x} - \boldsymbol{x_b})^T \boldsymbol{B^{-1}}(\boldsymbol{x} - \boldsymbol{x_b}) \qquad (1)$$

$J_o$ and $J_b$ are the cost functions calculated from observation terms and the model background. The observed value $\boldsymbol{y}$ ($= H(\boldsymbol{x})$)
is derived from the observation operator ($H$) and the observed input value $\boldsymbol{y_0}$, and the analysis field $\boldsymbol{x}$ is improved through DA
with the initial background field value $\boldsymbol{x_b}$. The covariance matrices $\boldsymbol{R}$ and $\boldsymbol{B}$ are from the observed error statistics and regional
background. $\boldsymbol{B}$ is calculated using the National Meteorological Center (NMC) method (Parrish and Derber, 1992), considering
12-h and 24-h forecasts of winter periods (Dec 01, 2017–Mar 31, 2018) with five members of control variables (u, v wind
component, surface pressure, temperature, and pseudo-relative humidity).

#### 2.2.2 LETKF DA method

For DA comparison, the WRF-LETKF system (Tsai et al. 2014) is applied, based on the concept of Hunt et al. (2007). The
system comprises a two-step process: analysis and forecast. The WRF model is used for forecasting during the cycling period.
For analysis, the system will update the prognostic variables, such as 3D wind, hydrometeor mixing ratio, potential temperature,
and geopotential height, using their BE covariances. Besides, the variable-depend horizontal and vertical localization radii are





set (Table 1). The observation errors for radial wind and reflectivity are assumed as 3 m s$^{-1}$ and 5 dBZ. Section 2.2.3 describes the observation operator method.

### 2.2.3 Radar observation operator

The model wind variable is converted to the Doppler radial velocity in 3DVAR and LETKF to assimilate the Doppler radar radial velocity ($V_r$).

$$V_r = u \frac{x-x_i}{r_i} + v \frac{y-y_i}{r_i} + (w - V_T) \frac{z-z_i}{r_i} \qquad (2)$$

Here, $x$, $y$, and $z$ are the radar locations, $x_i$, $y_i$, and $z_i$ are the radar observation data locations, $r_i$ [m] is the distance between the radar and the observation point, $u$, $v$, and $w$ [m s$^{-1}$] are the model wind components, and $V_T$ [m s$^{-1}$] is the terminal velocity.

The terminal velocity ($V_T$) can be related to rainwater mixing ratio ($q_r$) using Sun and Crook's (1997) eq. (3):

$$V_T = 5.40 a q_r^{0.125} \qquad (3)$$

where, $a$ is correctional term calculated by eq. (4).

$$a = \left( \frac{p_0}{\bar{p}} \right)^{0.4} \qquad (4)$$

$p_0$ is the ground pressure, and $\bar{p}$ is the average pressure [Pa]. The precipitating echo operator discussed in You et al. (2020)
is used for LETKF and that of Lee et al. (2020) for 3DVAR to assimilate reflectivity. Because the radar does not directly observe water vapor, the 3DVAR method improves the water vapor amount and temperature through assumptions based on the empirical relation between relative humidity and reflectivity (Wang et al., 2013). The water vapor amount ($q_v$) is assumed to be saturated ($q_s$) when the reflectivity observed is greater than 30 dBZ to simulate an environment in which convective clouds are actively developed and sustained in the atmosphere. The temperature of an air parcel is changed using the derived
relationship below.

$$q_v = 100\% \times q_s \qquad (5)$$

$$dq_v \cong q_s \times dRH + \frac{17.67 \times 243.5}{(T+243.5)^2} q_v dT \qquad (6)$$

RH is relative humidity and $T$ is temperature which are control variables improved through DA. Lee et al. (2022) improved the water vapor mixing ratio using reflectivity and improved the precipitation prediction accuracy. However, LETKF RDA
does not have an observation operator for water vapor and only improves the water vapor amount and temperature by calculating the covariance between the variables of the ensemble members.





### 2.2.4 Model configuration and experiment design

WRF version 4.1.3 (ARW; Skamarock et al. 2008) developed by NCAR of the University Corporation for Atmospheric Research is used in this study. The model domain consists of two nest grids with resolutions of 9 km (domain #1) and 3 km
(domain #2) centered over the Korean Peninsula (Fig. 3). Precipitation analysis was performed in the two regions: on the upstream region of Pyeongchang, GWD, where the center of the low-pressure system passed (red box area) and the main Olympic venue in Pyeongchang (blue box area).

This study used the rapid radiative transfer model (RRTM, Mlawer et al., 1997) for longwave radiation and the Dudhia scheme for short-wave radiation (Dudhia, 1989). The WRF double moment six-class scheme (WDM6; Lim et al., 2010) was
used for grid scale microphysical processes. The Kain–Fritsch cumulus parameterization scheme was applied to domain #1 and was turned-off for domain #2 (Kain 2004; Kain and Fritsch, 1990). The Yonsei University planetary boundary layer scheme (YSU PBL; Hong et al., 2006), the revised MM5 Monin–Obukhov surface process scheme, and the unified Noah land surface process (Jimenez et al., 2012, Tewari et al., 2004) were used in addition to aforementioned model physics. The National Center for Environmental Prediction (NCEP) Final Analysis (FNL) 1° × 1° data was used for obtaining the model's initial
condition and boundary condition (I.C. and B.C.).

To evaluate RDA effectiveness and precipitation forecast accuracy, two experiment groups were designed according to DA methods. The simulation experiment, where only the I.C. and B.C. were obtained without conducting DA, was referred as CTRL, making a 12–24-h prediction. In addition, the experiment conducting RDA using the LETKF DA method was referred to as LETKF and the experiment using the 3DVAR method was referred to as 3DVAR. The DA experimental group performed
RDA to domain #2 every 30 min for 3 h from the start. RDA was conducted seven times before creating a 12–24-h prediction field (Fig. 4). In the 3DVAR experiment, RDA is performed by setting CTRL as the initial field at the start of the DA period, but LETKF generates 30 ensembles the first time and predicts each ensemble until the first DA (Table 2). LETKF DA is performed for each ensemble member during cycling, averaging the ensemble analysis at the beginning of the forecast period and then performing one forecast. The details of the LETKF DA strategy can be found in You et al. (2020).


### 2.3 Overview of the cases

In Case 1, at 0000 UTC on December 23, 2017, a low pressure developed in southern China and moved to the Korean Peninsula, providing ample moisture. As the center of the cyclone passed through southern Korean Peninsula (Fig. 5 (a)), 13.4-cm snow was recorded in Daegwallyeong, and precipitation up to 24.8 mm was recorded in the red boxed area from 0000 UTC
to 1200 UTC on December 24, 2017.

In Case 2, similar to Case 1, the low pressure which developed over the East China Sea moved to the Korean Peninsula, and snowfall occurred. Due to this low pressure and high pressure in northern China, the eastern coast of the peninsula experienced strong easterly winds at 2100 UTC on March 18, 2018 (Fig. 5 (b)). In total, 8.0-cm snowfall was observed in Yongpyong, and





a maximum of 17.4-mm precipitation was recorded in the blue boxed area, and a maximum of 64.5-mm precipitation was
recorded in the red boxed area from 1200 UTC on March 18 to 0000 UTC March 19, 2018.

### 2.4 Verification statistics

We considered several verification parameters to objectively compare the model's prediction improvement. The quantitative
error statistics were calculated by comparing the 10-12 h accumulative precipitation from the forecast field and AWS data.
Equations (7) and (8) show the bias and root mean square error (RMSE).

$$\text{Bias} = \frac{1}{N}\sum_{i=1}^{N}(P_i - O_i) \tag{7}$$

and

$$\text{RMSE} = \sqrt{\frac{1}{N}\sum_{i=1}^{N}(P_i - O_i)^2}, \tag{8}$$

where $N$ is the number of horizontal grid points, $P_i$ is the precipitation (mm) in the model's predicted field, and $O_i$ is the
precipitation observed using AWS (mm).

The fractions skill score, proposed in Roberts and Lean (2008), illustrates a new method for forecast validation. A distance
would be set so that the forecast results could be acceptable for some position error, such as rainfall or reflectivity patterns.

$$FSS = 1 - \frac{\frac{1}{N}\sum_{N}(P_i - O_i)^2}{\frac{1}{N}(\sum_{N}P_i^2 + \sum_{N}O_i^2)} \tag{9}$$

The spatial correlation coefficient (SCC) presents a correlation between the predicted and observed precipitation distributions.
While the SCC value approaches 1, the accumulated rainfall's pattern and distribution are more synchronized. $\bar{P}$ is the mean
precipitation (mm) in the model's predicted field, and $\bar{O}$ is the mean precipitation observed using AWS (mm).

$$SCC = \sum \frac{\sum_{i=1}^{N}(P_i - \bar{P})(O_i - \bar{O})}{\sqrt{\sum_{i=1}^{N}(P_i - \bar{P})^2 \sum_{i=1}^{N}(O_i - \bar{O})^2}} \tag{10}$$

## 3 Results

### 3.1 Increment of the analysis field

Figure 6 shows the radar reflectivity, wind, snow, water vapor amount, and temperature increments at 3 km for 0000 UTC,
December 24, 2017. Figs. 6(b)–(e) show the difference between LETKF and CTRL, Figs. 6(f)–(i) show the difference between
3DVAR and CTRL, and Figs. 6(j)–(m) show the difference between LETKF and 3DVAR. Increments in wind and





hydrometeors show similar patterns, depending on the DA method. A difference occurs in each variable's increment. In
3DVAR, the snow amount in the southwest of the Korean Peninsula reduced by ~0.6 g kg$^{-1}$ and in LETKF by ~0.4 g kg$^{-1}$.
However, the water vapor amount and temperature increment pattern showed different trends, depending on the DA method.
The water vapor amount and temperature decreased where a reflectivity of 0 dBZ or more was observed in LETKF but
increased in 3DVAR because of the difference in water vapor and temperature assimilation between LETKF and 3DVAR
(Section 2.2.3). This difference in increasing trend resulted in a larger water vapor amount and temperature change in 3DVAR
and an underestimation in LETKF.

Case 2 showed the same trend as Case 1 (Fig. 7). The increase in the wind and snow mixing ratio is similar, regardless of the
DA method, and the snow mixing ratio increased by 0.2 g kg$^{-1}$ where a reflectivity of 35 dBZ or more was observed in LETKF,
but the increase was not evident in 3DVAR. The snow mixing ratio is higher in LETKF. Where a reflectivity of 15 dBZ or
more was observed, the inland water vapor amount and temperature decreased in LETKF and increased in 3DVAR. In LETKF,
a band-shaped decrement appears near North Korea. The water vapor amount in the background ensemble mean differs from
CTRL due to errors during ensemble forecasting.

### 3.2 Analysis field verification

DA methods were compared with KMA and ICE-POP radiosonde data to confirm which one simulated the water vapor
amount and temperature most similar to the observed atmosphere. Figure 8 shows the radiosonde profiles (black line) of OSAN
and MOO and the LETKF (red line) and 3DVAR (blue line) profiles at each observation point. The OSAN radiosonde data
are located in the western Korean Peninsula near the center of a low pressure and MOO is located in GWD. In Case 1, the
difference in the vertical profile from the ground to 500 hPa according to the DA method was large in OSAN, where the low
pressure was in the southeast of the Korean Peninsula. 3DVAR simulated a more humid atmosphere than observed below 850
hPa but showed a profile similar from 850 hPa to 400 hPa. LETKF underestimates temperature and dew point in the region
below 500 hPa. Because the interval between temperature and dew point temperature is wider than observed, a dryer
atmosphere is simulated. This is because reflectivity of more than 30dBZ is observed in this area and the water vapor amount
and temperature increase in 3DVAR, but decrease according to the covariance between variables in LETKF (Fig. 6). At MOO,
3DVAR simulated atmospheric profiles similar to those observed, compared to LETKF. In Case 2, at altitudes below 700 hPa
at all two sites, the LETKF underestimates the temperature, The atmospheric humidity of LETKF is similar to that of observed
from 700 to 800 hPa, but because the temperature is lower than the observed, mixing ratio of water vapor is relatively small.
A relatively dry area exist from ground to 900 hPa, and this area is also simulated more accurately by 3DVAR than LETKF.

Quantitative verification was performed to accurately verify the numerical results. Error statistics were performed for air
temperature below 500 hPa and the dew point temperature, and the wind at 850 hPa, 700 hPa, and 500 hPa, respectively. In
the temperature and dew point temperature verification, 3DVAR showed a low error (Table 3). In LETKF, negative
temperature biases were derived from all sites, and the dew point temperature was simulated lower than observed at the OSAN





site, where the low pressure was located. In other words, the air in the southern Korean Peninsula, where the low pressure was in the LETKF was colder and dryer. However, in the wind verification results, the low wind error was calculated in LETKF, contrary to the temperature and dew point temperature verification results (Table 4). The LETKF showed high accuracy in the

case of wind because the radial velocity was assimilated considering the real time model error. However, the accuracy of the water vapor amount and temperature was lowered using the covariance between ensemble models without using radar information.

### 3.3 Analysis of hydrometeor mixing ratio vertical profile

Figure 9 shows the difference in the amount of water vapor and hydrometeor mixing ratio vertical profile in the analysis field

between experiments. (a), (d) is the difference between LETKF and CTRL, (b), (e) between 3DVAR and CTRL, and (c), (f) between LETKF-3DVAR. At 0000 UTC, December 24, 2017, the water vapor amount increased by 0.02 g kg$^{-1}$ in some sections (1.5–3 km, 5–7 km) in the LETKF experiment, whereas in 3DVAR, it increased to 0.06 g kg$^{-1}$ up to 7 km (Fig. 9(a), (b)). Note the amount of change in the snow mixing ratio. In LETKF, it decreased by 0.14 g kg$^{-1}$, compared to CTRL, but in 3DVAR, it decreased by 0.22 g kg$^{-1}$. Compared to 3DVAR, LETKF underestimated the water vapor and over simulated the

snow amount (Fig. 9(c)). In Case 2, the difference in water vapor amount was more pronounced than in Case 1. Compared with CTRL in LETKF, the water vapor amount reduced to an altitude of 8 km or less (Fig. 9(d)). The decrease in the water vapor amount increased to a maximum of 0.13 g kg$^{-1}$. However, 3DVAR increased the water vapor amount at an altitude of 3 km or less, compared to CTRL (Fig. 9(e)). Like Case 1, LETKF simulated a smaller amount of water vapor mixing ratio and a larger amount of snow mixing ratio than 3DVAR (Fig. 9(f)).

Figures 10 and 11 show the water vapor, snow, rain mixing ratio time height section of CTRL, 3DVAR, and LETKF for the blue region during the forecast period. In Case 1, the snow mixing ratio of 3DVAR increases to 0.3g kg$^{-1}$ at 0300 UTC, but LETKF simulates only half the amount during the forecast period (Fig. 10(b), (e) and (h)). In figure 9 (c), the amount of snow mixing ratio in the LETKF analysis field was larger than 3DVAR at an altitude of 2 km or higher. However, in the prediction field, the opposite trend is observed. Because the water vapor amount of LETKF decreased and the atmosphere became dry,

hydrometeors were not grown or formed during the forecast period compared to 3DVAR (Fig. 10(d)-(f)). Rain mixing ratios were also simulated the most in 3DVAR and the least in LETKF (Fig. 10(f), (i)). At 1700 UTC in Case 2, a significant increase occurred in the 3DVAR mixing ratio of snow and rain, compared to CTRL and LETKF (Fig. 11(c), (f) and (i)). Hydrometeors formed through the microphysical process by simulating sufficient water vapor (Fig. 11(g)-(i)). LETKF simulated the lowest water vapor mixing ratio, and only a few hydrometeors formed during the forecast period (Fig. 11(d)-(f)). The LETKF analysis

field had a higher snow mixing ratio than 3DVAR, but in the prediction field, a larger snow mixing ratio occurred in 3DVAR. Therefore, the water vapor mixing ratio significantly influences snowfall precipitation and its formation.





### 3.4 Radar reflectivity contoured frequency by altitude diagram (CFAD)

The Korean Peninsula has mountainous terrain so it is not suitable to locate radars near the ground due to beam blockage.
For this reason, most radars are located on top of the mountain and because radar observes in a cone-shaped way, the number
of low-level data is reduced when generating the CAPPI data. CFAD provides information of frequency distribution for a
variable in a given radar reflectivity volume in a single contour plot (Yuter and Houze 1995). Frequency distribution is
displayed as a percentage (%) by normalizing radar data by altitude. Figure 12 (a) and (d) is a CFAD, showing the reflectivity
within a radius of 100 km of the GRS radar in GWD (blue region). The observed GRS radar CFAD shows that the reflectivity
increases from 10 km for Case 1, indicating that the hydrometeors formed from above 10 km and slowly grew as they reached
the ground (Fig. 12(a)). In LETKF, the hydrometeors grew rapidly from ~6 km, lower than observed. In LETKF, the solid
hydrometeors grew to an average of 27 dBZ, but in 3DVAR, they grew to solid hydrometeors showing higher reflectivity at
~5 dBZ (Fig. 12(b), (c)). The 3DVAR starts at a higher altitude than LETKF. From the radiosonde profile, the hydrometeor
growth layer is also thin because the simulated atmosphere in LETKF is dry (Fig. 8(b)). 3DVAR simulated an increase in
reflectivity from 10 km to low altitudes, similar to observations (Fig. 12(c)). In Case 2, solid hydrometeor formation in GRS
and 3DVAR starts at 10 km as in Case 1(Fig. 12 (d), (f)). However, in LETKF, solid hydrometeors are generated at 7 km
(Fig. 12 (e)). In LETKF, solid hydrometeors grow on an average of 25 dBZ, whereas 3DVAR grows to 30 dBZ.

As shown in the Skew-T Log-P diagrams, the water vapor amount increased through RDA in 3DVAR, and the atmosphere
was more humid than that of LETKF (Fig. 8). This allowed water vapor to grow into hydrometeors from the upper troposphere.
In  Case 2 CFAD of GRS (Fig. 12 (d)), there is a shallow layer of high reflectivity frequency near the ground which indicates
hydrometeor growth due to advection of moisture from the ocean for both cases. However, in the model prediction, the
reflectivity rapidly decreased below the melting layer due to evaporation of falling hydrometeors. This can be confirmed in
Figure 11 (c), (f), (i) as the rain mixing ratio peaks at about 1.5 km, not at the ground.

### 3.5 Distribution of cumulative precipitation and verification

Figure 13 shows the 12-hour cumulative predicted precipitation distribution. In Case 1, The snowfall in GWD was less
simulated in LETKF, whereas snowfall of 10.0 mm or more was simulated in 3DVAR, showing a precipitation pattern similar
to the observation (Fig. 13(a)-(c)). A negative bias of −6.031mm was derived by verifying the cumulative precipitation error
of LETKF, underestimating precipitation (Table 5). Similar to Case 1, in Case 2, the water vapor amount decreased through
LETKF DA, and precipitation was underestimated (Fig. 13(d)-(e)). The LETKF showed a larger negative deviation than
3DVAR, and the RMSE was calculated to be 2.238 mm higher (Table 5). In the 3DVAR experiment, Case 1 and Case 2 over
simulated precipitation in the southern Korean Peninsula, where the low-pressure center passed. Because the water vapor
mixing ratio observation operator was empirically formulated in 3DVAR, the water vapor mixing ratio was inputted
excessively and a simulation result of more precipitation than the actual observation occurred.





FSS score verification was performed for 60-minute precipitation in the entire South Korea (domain #2: all-region) using AWS data (Fig. 14). In Case 1, the 3DVAR FSS score showed a high value of 0.6 or more in all and blue regions from 2 to 4

hour prediction. However, the accuracy of 3DVAR decreases after 4 hours, and LETKF showed higher FSS value than 3DVAR in the 8-hour prediction in the all-region and the 7-hour prediction in the blue region. At the beginning of the prediction, 3DVAR showed higher accuracy than LETKF because the water vapor mixing ratio assimilated in 3DVAR analysis field helped to form precipitation. However, after 6 hours, the 3DVAR winds became less accurate and did not crossover the Taebaek Mountains and stagnated in the valleys of Gangwon-do, increasing the precipitation error. In Case 2, 3DVAR showed

higher accuracy up to 6 hours of prediction in all regions. In the blue region, the FSS score of 3DVAR was lower than LETKF until 3 hours of prediction. Even in Case 1, the FSS score in the 1-hour prediction of 3DVAR was as low as 0.49. This can be accredited to the fact that hydrometeors mixing ratio of 3DVAR in the blue region decreased compared to CTRL and the increase in the water vapor amount was small compared to that in the red region in the analysis field. Thus, the hydrometeors were not sufficiently formed at the beginning of the prediction time. This can be confirmed in Figures 10 and 11 that the ratio

of 3DVAR and snow mixing ratio at the beginning of prediction was less than that of CTRL.

Precipitation statistics using AWS can verify precipitation observed only on land, but radar can be used as a proxy for precipitation over land and the ocean. So the SCC verification was performed using radar reflectivity at 2 km to avoid terrain blockage (Fig. 15). In Case 1, LETKF scored higher than 3DVAR up to 9 hours prediction. Although the precipitation of LETKF was underestimated, LETKF was able to simulate the distribution of precipitation, and showed a slightly higher SCC

value than 3DVAR. In addition, as shown in Table 4, the wind error of 3DVAR was larger than LETKF. During the forecast period, the error of the wind increased and the wind did not cross the Taebaek Mountains, so the inland air became stagnant, forming hydrometeors and precipitation in the wrong location, and a pattern different from the observed reflectivity appeared. In all region of Case 2, SCC showed a similar tendency as the FSS. Note that for precipitation evaluation, it is important to compare not only over the land but also over the ocean to have an accurate picture of the statistics.

**4 Summary and Conclusion**

This study compared two methods for assimilating radar data in a numerical weather prediction model using methods of LETKF and 3DVAR to improve the forecast accuracy of winter precipitation on the Korean peninsula with complex terrain. The effectiveness of RDA and forecast improvement according to two DA methods were verified by performing two experiments of heavy snowfall cases and analyzing the accumulated precipitation spatial patterns, hydrometeor profiles, and

cloud microphysical processes. KMA and MOE S-band radar data were assimilated and the results were evaluated using ICE-POP 2018 observations.

A numerical experiment performed using only the I.C. and B.C. without conducting DA was named CTRL. It was used to measure the degree of DA improvement among experiments. The LETKF experiment considered BEs of ensemble members, and the 3DVAR experiment considered only climatic BEs. LETKF improved the water vapor amount and temperature using





the covariance of the ensemble members, but 3DVAR improved the water vapor amount and temperature through an operator that assumed the atmosphere was saturated when reflectivity was above a certain threshold.

Increments in wind and hydrometeors showed similar patterns for both DA methods but subtle differences were found. In 3DVAR, the snow amount in the southwest of the Korean Peninsula was reduced compared to LETKF. Temperature and the water vapor amount decreased where a reflectivity of 0 dBZ or more was observed in LETKF but increased in 3DVAR because

of the difference in the assimilation methods. This difference in trend resulted in larger amounts of water vapor mixing ratio and temperature change in 3DVAR and an underestimation in LETKF. From the analysis field verification using ICE-POP observations, wind in LETKF was more accurately simulated than 3DVAR but underestimated the water vapor mixing ratio and temperature in the lower troposphere due to a lack of a water vapor and temperature observation operator. In LETKF experiment, a dry atmosphere was simulated, compared to 3DVAR experiment, and a few hydrometeors were simulated during

the prediction period. Snowfall in GWD was less simulated in LETKF, whereas snowfall of 10.0 mm or more was simulated in 3DVAR, showing a higher accuracy than LETKF experiment, resulting in an error of 2.62 mm lower than LETKF experiment. The hourly rainfall FSS statistics in the entire region of South Korea showed that 3DVARhad a higher skill score up to 6 hours of the forecast but then the accuracy decreased. The wind error gradually increased and the precipitation error increased. In Case 1, LETKF showed higher accuracy in reflectivity SCC verification. This indicates that there is a gap in

reflectivity-precipitation relationship in the model. In addition, for precipitation evaluation, it is important to compare not only over the land but also over the ocean to have an accurate picture of the verification statistics. RDA of LETKF can be further improved by applying the water vapor operator similar to 3DVAR by forming more hydrometeors in clouds. In Case 2, a band-shaped increment appeared near North Korea due to a prediction error during ensemble prediction. Although the initial conditions of ensemble members are perturbed by the initial conditions of a deterministic forecast, prediction errors can

increase the system's inconsistency, resulting in different water vapor distributions among ensemble members.

The water vapor mixing ratio and temperature DA method performed using 3DVAR significantly affected precipitation prediction improvement. Therefore, it is necessary to apply observation operators for water vapor mixing ratio and temperature, even when assimilating LETKF data. However, existing methods for adjusting water vapor mixing ratios are rudimentary and empirical. 3DVAR over simulated precipitation in the southern part of the Korean Peninsula and requires adjustment in the

operator. Research is underway to develop a relationship between reflectivity and water vapor amount suitable for winter in Korea and the surrounding regions. Furthermore, the effect of the water vapor amount on snowfall should be studied in detail through cloud microphysical budget analysis. Wind is also vital in snowfall formation in GWD, and hydrometeor advection cannot be ignored. We plan to report detailed wind analysis during the forecast period and its impact on precipitation using the ICE-POP observation network.


**Code and data availability**

The original WRF and WRFDA code can be downloaded via the online repository at https://github.com/wrf-model/WRF/releases. The copyright of the original code for LETKF belongs to Takemasa Miyoshi, and it can be accessed from https://github.com/takemasa-miyoshi/letkf. National Center for Environmental Prediction (NCEP) Final Analysis (FNL)

$1° \times 1°$ data for initial and boundary conditions is available at https://rda.ucar.edu/datasets/ds083.2/. Observation data from KMA are available online (https://data.kma.go.kr) and ICE-POP data are available via https://doi.org/10.1594/PANGAEA.918315. The model codes and scripts that cover data and figure processing action for all of the results reported in this paper are available at https://zenodo.org/record/6378081#.YjryRedBwuU. Model outputs are available upon request to Ji-Won Lee (leejiwon2040@knu.ac.kr).

**Author contribution**

Ki-Hong Min conceptualized the paper, wrote the original draft, supervised the project, performed analysis, and provided resources; Kao-Shen Chung co-conceptualized the paper, administered the project, performed analysis, and reviewed and edited the paper; Ji-Won Lee and Cheng-Rong You assisted with data curation, conducted the numerical experiments and performed analysis and visualization of the research; Gyuwon Lee provided the ICE-POP data, administered the project, and

reviewed and edited the paper.

**Competing interests**

The authors declare that they have no known competing interests or personal relationships that could have appeared to influence the work reported in this paper.

**Acknowledgements**

This work was supported by the National Research Foundation of Korea (NRF) grant funded by the Korea government (MSIT) (2021R1A4A1032646). This work was also funded by the Korea Meteorological Administration Research and Development Program under grant KMIPA 2017-7010.




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





**Table 1.** Horizontal and vertical localization radii setting of each prognostic model's variables in the weather research and forecasting (WRF)-local ensemble transform Kalman filter (LETKF) data assimilation (DA) system.

| Prognostic Variables | U & V | W\PH\T | $q_v\backslash q_c\backslash q_i$ | $q_r\backslash q_s\backslash q_g$ |
|---|---|---|---|---|
| horizontal localization radius (km) | 36 | 12 | 24 | 12 |
| vertical localization radius (km) | | | 4 | |

**Table 2.** Selected cases for numerical experiments and their storm characteristics.

| | | | | | Maximum accumulated precipitation | |
|---|---|---|---|---|---|---|
| | Characteristics | Total period | Data assimilation (DA) period | Forecast period | Southern part of the Korean Peninsula (red box) | Pyeongchang region (blue box), |
| Case 1 | Shallow precipitation system, (Warm Low*) | 0000 UTC, Dec 23, 2017–0000 UTC, Dec 25, 2017 | 2100 UTC, Dec 23, 2017–0000 UTC, Dec 24, 2017 | 0000 UTC, Dec 24, 2017–0000 UTC, Dec 25, 2017 | 24.8 mm | 13.4 mm |
| Case 2 | Shallow precipitation system, (Warm Low*) | 0000 UTC, Mar 18, 2018–0000 UTC, Mar 20, 2018 | 0900–12000 UTC, Mar 18, 2018 | 1200 UTC, Mar 18, 2018–0000 UTC, Mar 20, 2018 | 64.5 mm | 17.4 mm |

* As defined in Kim et al. (2021).


**Table 3.** Temperature and dew point temperature verification using radiosonde for Case 1 (0000 UTC, December 24, 2017) and Case 2 (1200 UTC, March 18, 2018).

| Bias | | Experiment | OSAN | MOO |
|---|---|---|---|---|
| Temperature [K] | Case 1 | LETKF | −0.789 | −0.623 |
| | | 3DVAR | 0.071 | −0.168 |
| | Case 2 | LETKF | −2.280 | −0.699 |
| | | 3DVAR | −1.086 | −0.614 |
| Dew point temperature [K] | Case 1 | LETKF | −3.756 | 2.435 |
| | | 3DVAR | 1.949 | 0.521 |
| | Case 2 | LETKF | −0.655 | 1.511 |
| | | 3DVAR | −0.221 | 1.068 |





**Table 4.** Wind verification using radiosonde for Case 1 (0000 UTC, December 24, 2017) and Case 2 (1200 UTC, March 18, 2018).

| Wind RMSE [m/s] | | 850 hPa | 700 hPa | 500 hPa | Average |
|---|---|---|---|---|---|
| Case 1 | LETKF | 8.575 | 5.701 | 9.240 | 7.839 |
| | 3DVAR | 9.044 | 7.208 | 9.981 | 8.744 |
| Case 2 | LETKF | 8.418 | 7.418 | 11.798 | 9.211 |
| | 3DVAR | 7.850 | 7.780 | 13.218 | 9.616 |


**Table 5.** 12-hr cumulative precipitation bias, RMSE for LETKF, 3DVAR in blue area.

| | Experiment | Bias [mm] | RMSE [mm] |
|---|---|---|---|
| **Case 1** | LETKF | −6.031 | 6.924 |
| | 3DVAR | 0.608 | 3.925 |
| **Case 2** | LETKF | −3.819 | 4.949 |
| | 3DVAR | −0.838 | 2.711 |



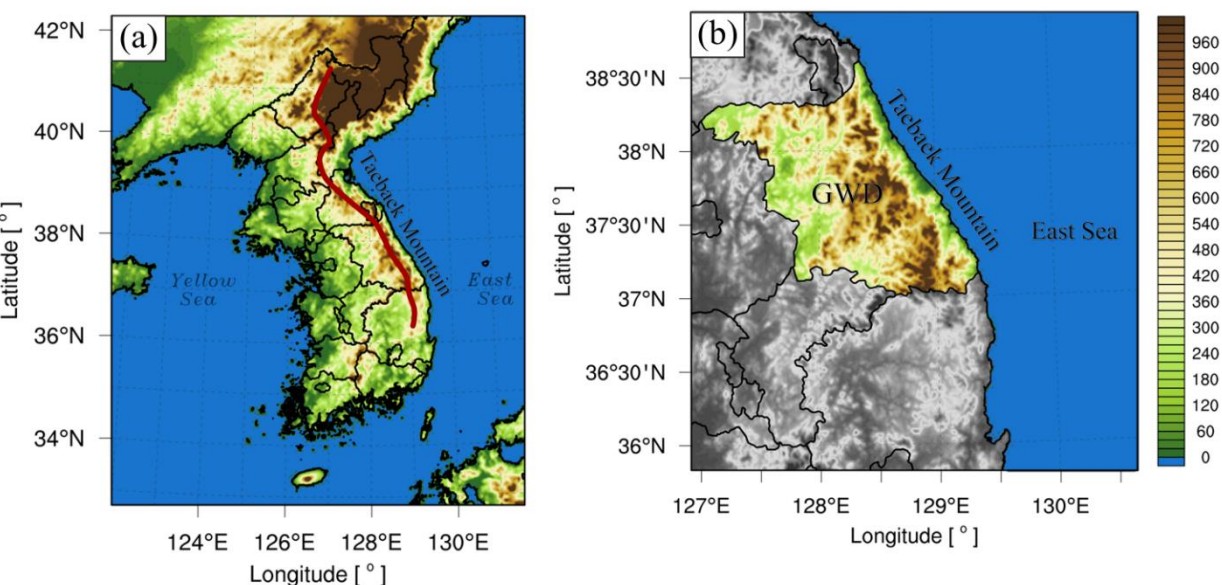

**Figure 1: (a) Korean Peninsula and (b) Gangwon-do (GWD) topography and topographic altitude [m]. The red line is the Taebaek Mountains.**

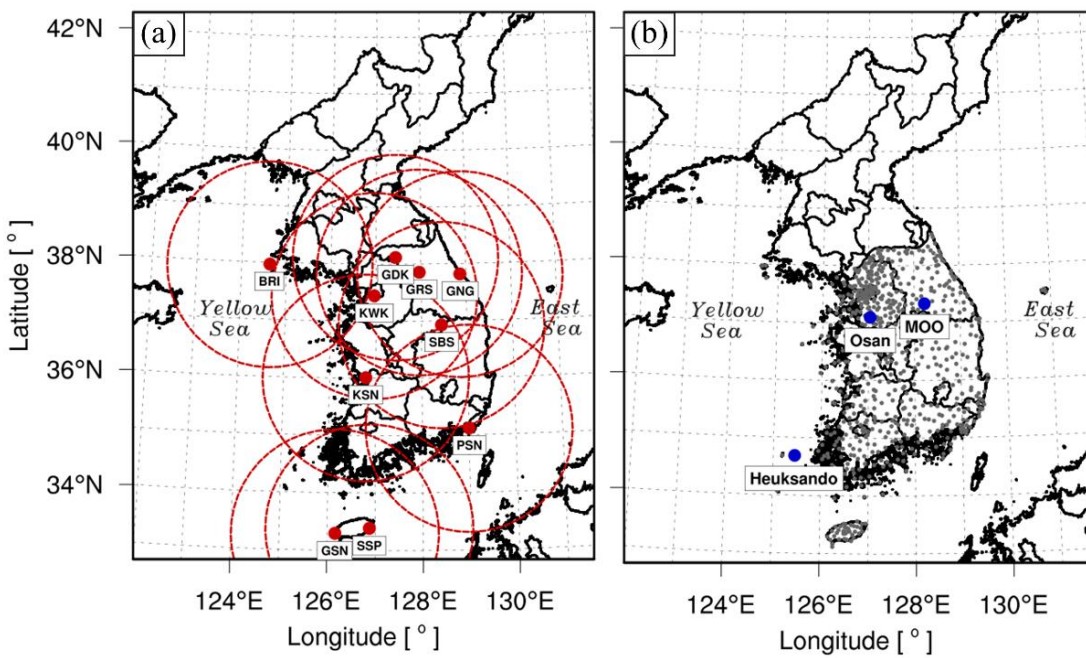

**Figure 2: Locations of (a) KMA and the MOE S-band radar sites (red dots) with the radar-coverage areas in circles and (b) AWSs**
**(gray dots) and radiosonde (blue dots). The radar site abbreviations are GDK: Gwangdeok Mountain; GSN: Gosan; KWK: Gwanak Mountain; PSN: Gudeok Mountain; SSP Seongsan; KSN: Oseong Mountain; BRI: Baengnyeongdo; SBS: Sobaek Mountain; GRS: Gari Mountain (GRS), repectively.**





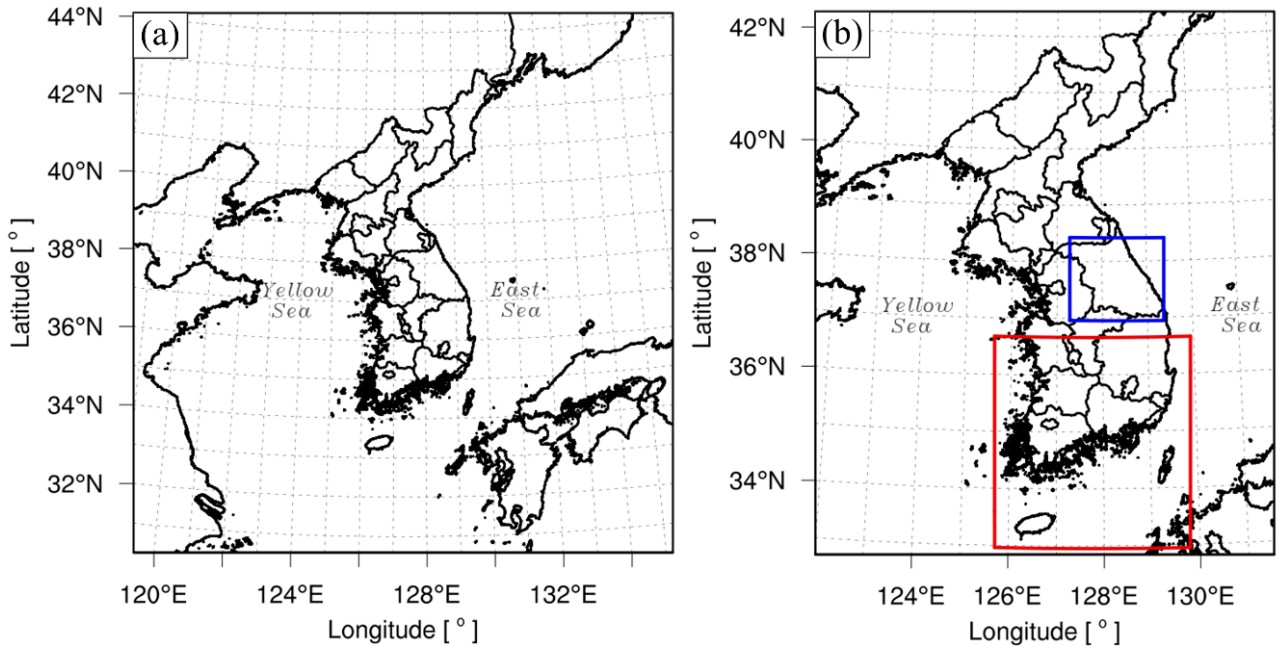

**Figure 3: (a) Domain 1, (b) domain 2 area and analysis domains (blue: Pyeongchang area; red: southern part of the Korean Peninsula).**

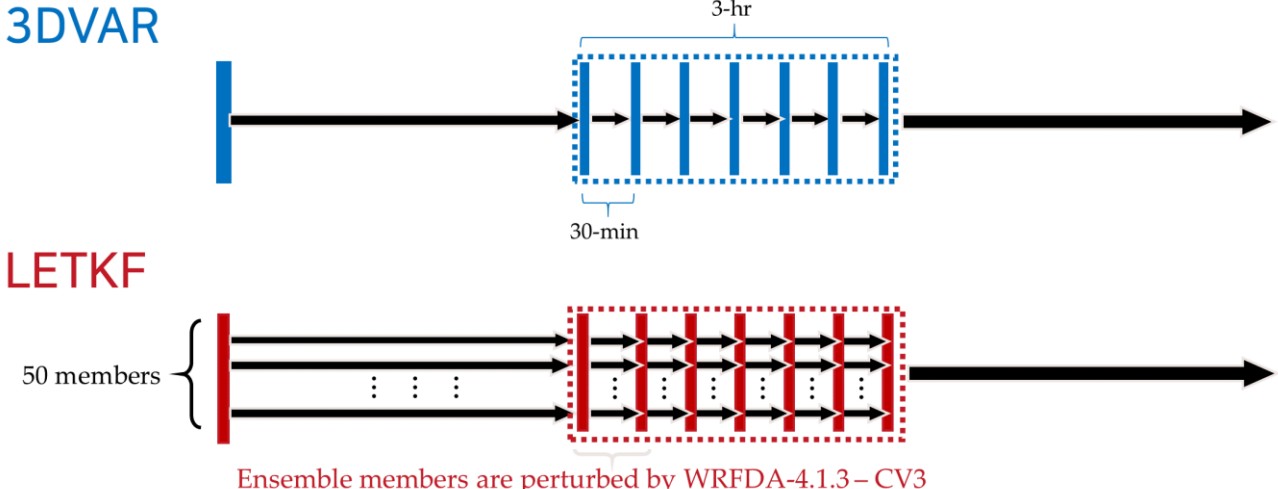

**Figure 4: Assimilation strategy of 3DVAR and local ensemble transform Kalman filter (LETKF) RDA methods.**





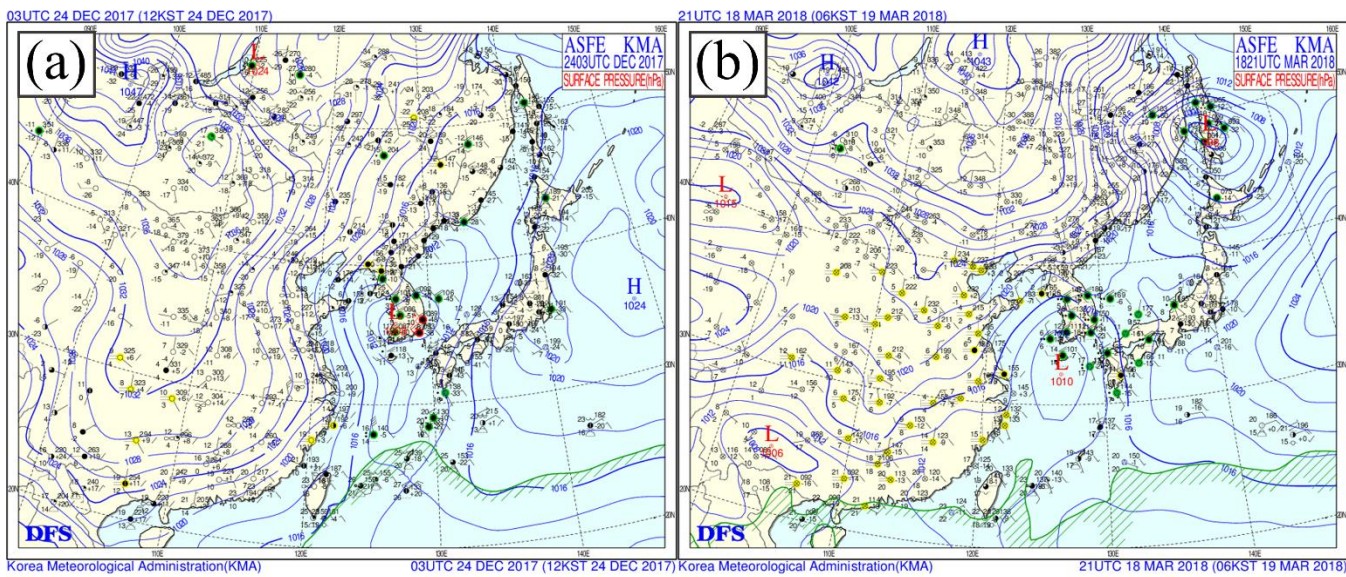

**Figure 5: The surface weather chart from KMA for (a) 0300 UTC, Dec 24, 2017, and (b) 2100 UTC, Mar 18, 2018.**



**Figure 6:** 3 km (a) radar reflectivity; (b), (f), and (j) wind; (c), (g), and (k) snow; (d), (h), and (l) water vapor mixing ratio; (e), (i), and (m) temperature between (b)–(e) LETKF and CTRL, (f)–(i) 3DVAR and CTRL, and (j)–(m) LETKF and 3DVAR at 0000 UTC, December 24, 2017, respectively.



**Figure 7: The same as Figure 6, except for 1200 UTC, March 18, 2018.**



**Figure 8: Skew-T Log-P diagrams in (a), (c) OSAN and (b), (d) MOO for radiosonde (black line), LETKF (red line) and 3DVAR**
**(blue line) for (a)–(b) 0000 UTC, December 24, 2017, (c)–(d) 1200 UTC, March 18, 2018, respectively.**



Figure 9: Vertical profiles of water vapour(black line) and five hydrometeor mixing ratios(cloud: red, rain: yellow, ice: green, snow: blue and graupel: purple line) differences between (a), (d) LETKF and CTRL, (b), (e) 3DVAR and CTRL, (c), (f) LETKF and 3DVAR for (a)–(c) 0000 UTC, December 24, 2017, (d)–(f) 1200 UTC, March 18, 2018, respectively.





Geoscientific Model Development Discussions — Open Access — EGU

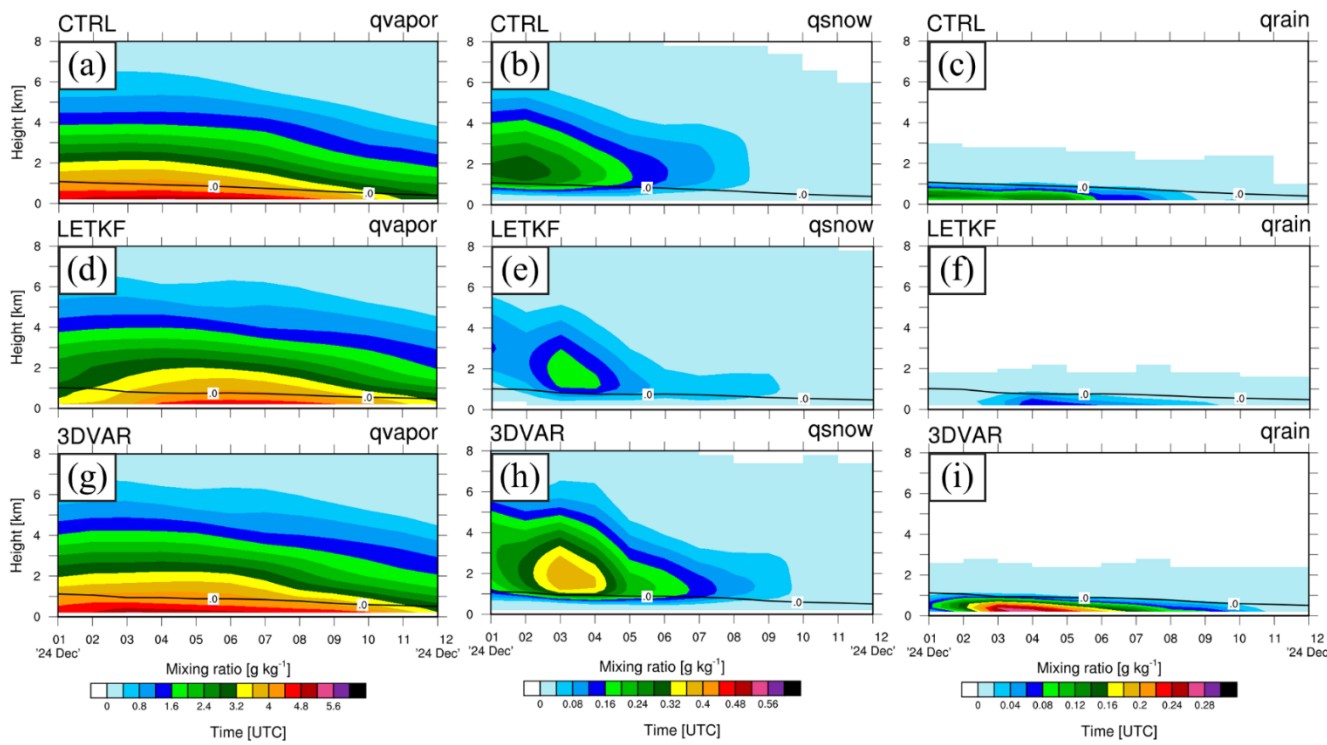


**Figure 10: (a), (d), and (g) water vapor; (b), (e), and (h) snow; and (c), (f), and (i) rain mixing ratio time height section of (a)–(c) CTRL, (d)–(f) LETKF, and (g)–(i) 3DVAR for the blue region of Case 1.**



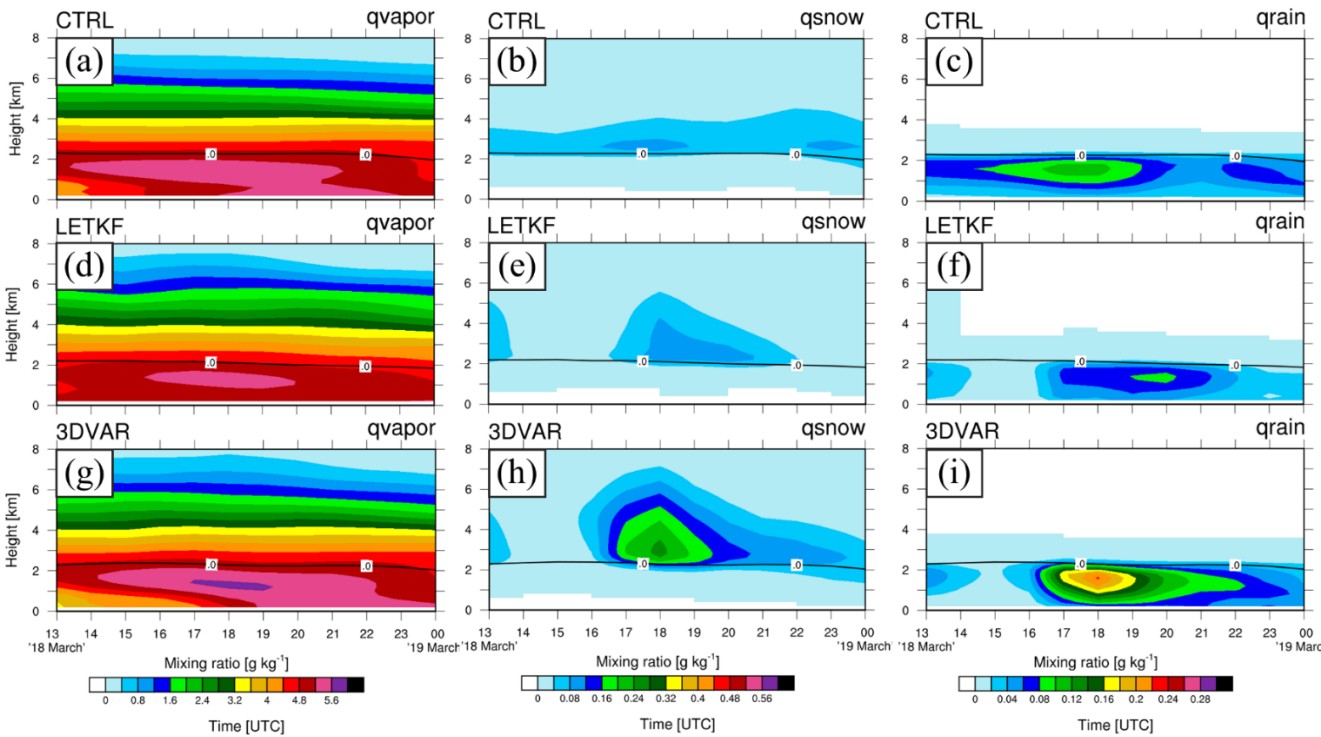

Figure 11: The same as Figure 10, except for Case 2.

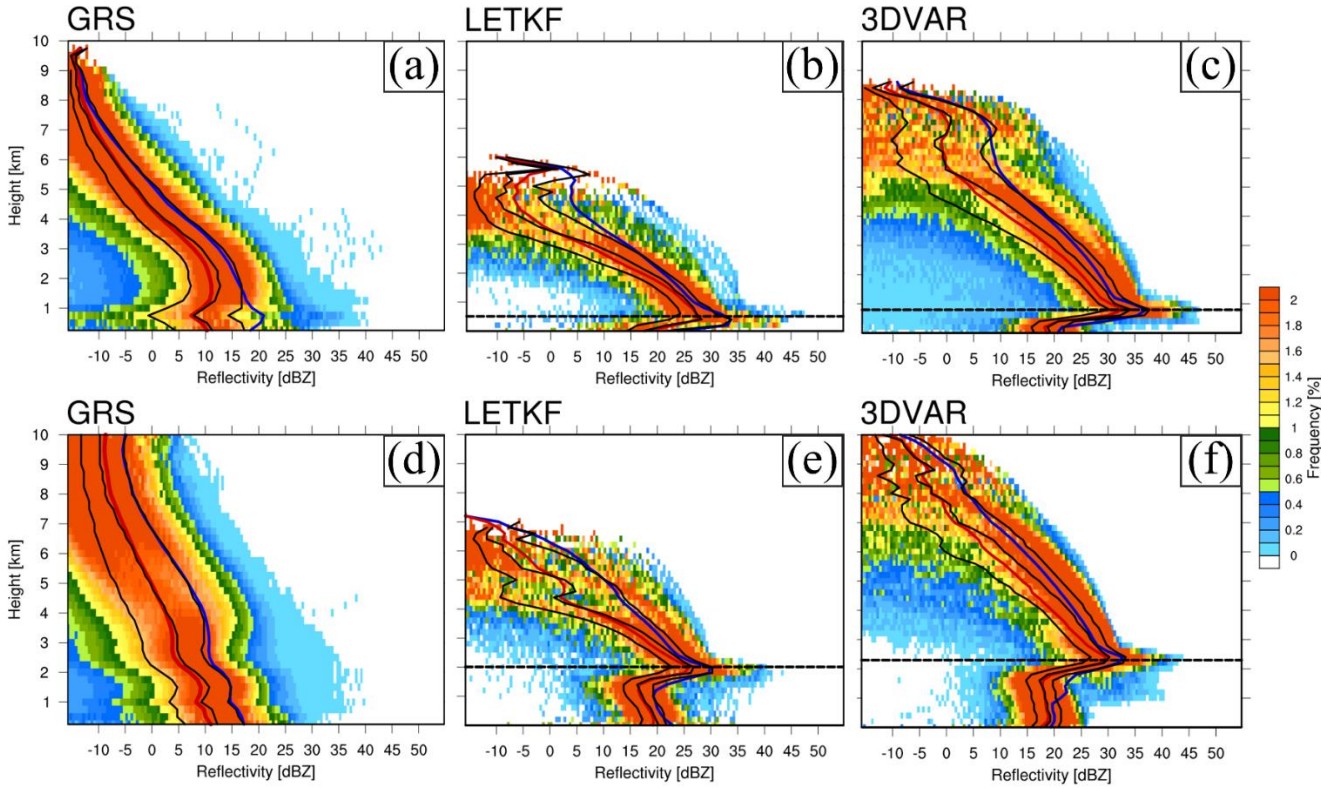

**Figure 12: Contoured frequency by altitude diagram (CFAD) for (a), (d) GRS observation and simulated radar reflectivity from (b), (e) LETKF and (c), (f) 3DVAR. The cumulative reflectivity frequencies of the 25th, 50th, and 75th percentiles are marked with black solid lines, the average reflectivity (dBZ) is marked by a red solid line, and the average reflectivity factor in linear units (mm$^{-6}$ m$^{-3}$) is marked by a blue solid line. The horizontal black dashed line in panel (b)–(f) represents the model's 0°C height for (a)–(c) Case 1 and (d)–(f) Case 2.**





**Figure 13: 12-hr cumulative precipitation (mm) distribution of (a)–(c) Case 1, (d)–(f) forecast period for (a), (d) AWS, (b), (e) LETKF, and (c), (f) 3DVAR.**




**Figure 14: 60-minute precipitation FSS for (a), (c) all region and (b), (d) blue region in (a)–(c) Case 1 and (b)–(d) Case 2 for LETKF (red line) and 3DVAR (blue line).**

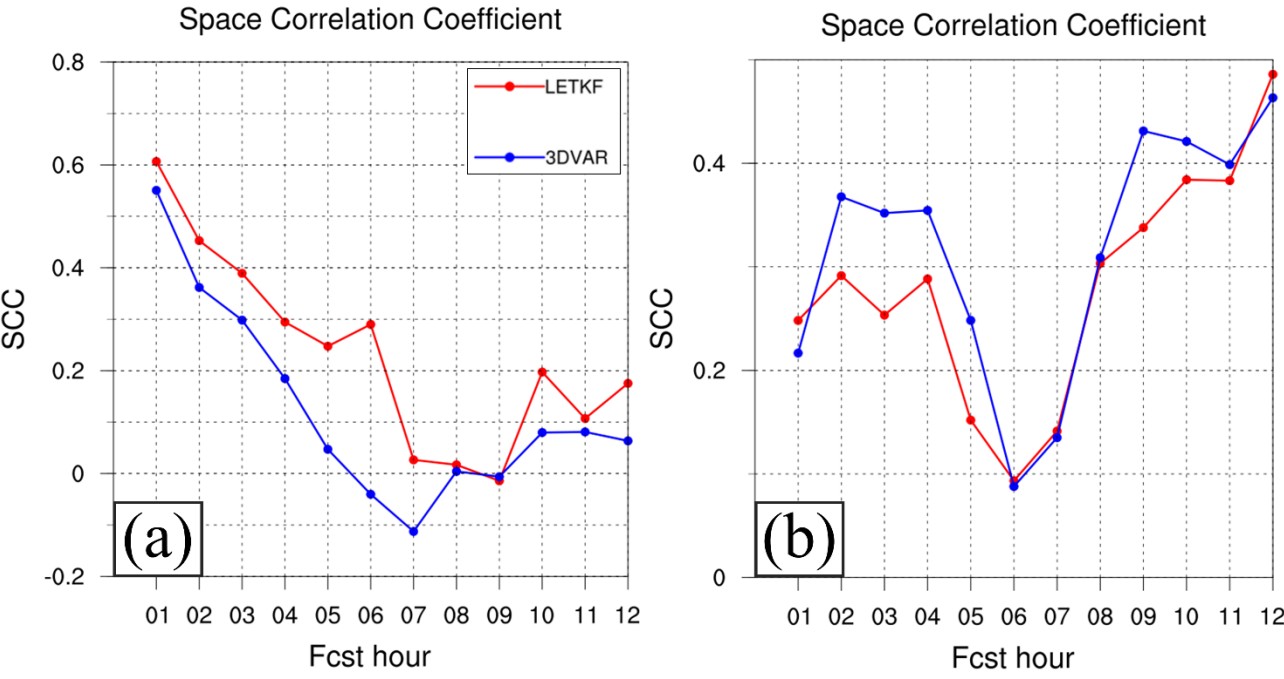

**Figure 15: 2 km reflectivity SCC verification values for all region in (a) Case 1 and (b) Case 2 for LETKF (red line) and 3DVAR**
**(blue line).**