# Peer review of "Intercomparing radar data assimilation systems for ICE-POP 2018 snowfall cases"

_Geoscientific Model Development, 2022_

## Referee Comment (RC2)

This study compares two radar data assimilation methods including 3DVAR and LETKF for ICE-POP snowfall cases. It was found that water vapor adjustment is important for the radar data assimilation. However, water vapor and temperature adjustment has been developed for years. What is new for this conclusion? Also, it is weird to me that the assimilation of radar observation results constant negative increment of snow mixing ratio. Detail descriptions of reflectivity observation operators are needed. A possible explanation is the current radar data assimilation does not adjust the hydrometer variables. In that case, what is the meaning of reflectivity assimilation? Overall, I'm expecting some improvements of data assimilation method. Simple comparison between LETKF and 3DVAR does not meet the main target of GMD. Therefore, I will not recommend this paper for a publication in GMD.

Some more specific points are listed below:

1. Line 15-20: "… a lack of a water vapor and temperature observation operator." What does this mean? I believe the radar data assimilation also adjusts water vapor and temperature in LETKF. The cross variable correlation is an advantage for the common used EnKF. On the contrary, the empirical saturation adjustment of 3DVRA will lead to serious overestimation of precipitation. Please see more in Carlin et al (2017).

2. For section 2.2.3, how is the reflectivity data assimilated? Any adjustment of hydrometer variables for 3DVAR and LETKF? Detail descriptions are needed. Most important, for snowfall radar data assimilation cases, I think the direct adjustment of snow mixing ration for radar data assimilation are necessary. Otherwise, any explanations of the improvement of snow are hardly convincible.

3. About the LETKF, what are the analysis variables?

4. Line 172: 30 members? But it shows 50 members in Figure 4. How the initial members generated?

5. The spinup period before the radar data assimilation?

6. Line 195-200: A distance? What value?

7. Line 208-209, "…, depending on the DA method". What does this mean?

8. For Figure 6, the final analysis or the first analysis? Any explanation why both LETKF and 3DVAR get negative increment of snow mixing ratio?

9. For Figure 9, in what region? Key information should be clearly stated in the figure caption.

10. Please rewrite section 3. Studies have already compared the difference between 3DVAR and LETKF. What is new for this comparison?

---

## Referee Comment (RC3)

**Comment on gmd-2022-18**

Referee comment on "Intercomparing radar data assimilation systems for ICE-POP 2018 snowfall cases" by Ki-Hong Min et al., Geosci. Model Dev. Discuss., https://doi.org/10.5194/gmd-2022-18-RC1, 2022

**General comments:**

The paper uses radar data to run two different data assimilation methods, LETKF and 3dVAR, during two snowfall events observed in ICE-POP 2018 fleld campaign. The authors compared the analysis and the forecasted data of different variables including wind, water vapor, temperature, and snowfall to show the importance of water vapor assimilation. The logic and structure of the paper are clear and easy to follow. However, several sentences need to be rewritten/reconsidered as well as some concerns which need to be considered before publication.

**Specific comments:**

- It seems the authors run only one forecast cycle to compare two methods which cause few samples to do the verification methods. Please consider running more forecast cycles to have enough samples which make the verification as well as the results more reasonable.
- The authors use two different data assimilation methods; however, in many sentences particularly in the results and summary sections, the "simulation" word was used to refer to the LETKF or 3DVAR methods. Please note that this is the wrong word referring to the assimilation method. The LETKF and 3DVAR do the "assimilation" not "simulation". Sentences like „The snowfall in GWD was less simulated in LETKF" are logically wrong and need to be rewritten.
- The word "underestimate" was used many times in sections 3 and 4 to compare two assimilation methods. Since non of these methods were considered as a reference experiment or reference data, using the words "underestimation" or "overestimation" for this comparison is meaning less. The words "underestimation" or "overestimation" could be used when the results are compared with reference data such as observation. Please consider rewriting these sentences.
- In section 2.2.2 the radial wind and the reflectivity errors are assumed 3 ms-1 and 5 dbz respectively; however, the authors did not mention the source of these numbers.
- In section 2.3 was mentioned that the precipitation up to 24.8 mm was recorded in the red box area from 00 to 12 UTC. It is an unclear sentence. The precipitation reported from an SYNOP station (which probably is the concern of the author in this sentence) would be for a specific point not for a whole specific area. The sentence could be rewritten by pointing to the minimum and maximum precipitation amount in this area as well as the location of the station which had a maximum report of precipitation. Please consider also the last paragraph in this section which has the same problem.

- In section 3.1 the sentence „The snow mixing ratio is higher in LETKF" is a general sentence that of course is not generally correct. Please consider mentioning clearly about the specific case (time/location) where the assimilated snow mixing ratio is higher than assimilated snow mixing ratio in 3DVAR.
- In Fig. 8, there is no explanation for the dashed lines.
- Fig. 14(d), there is a weird feature regarding the FSS score. The FSS scores for both LETKF and 3DVAR methods are very low at the first forecast hours and they increase after about 6 hours. This is not a common behavior in the FSS score of a forecast validation. It would be good if the authors recheck this case or explain a bit about such a weird behavior of the FSS score.

**Technical correction:**

- Line 35: „cool ocean winds" → „cold ocean winds"
- Line 49: „has" → „get"
- Line 69: „only include information → „include only information"
- Line88: „Further" → „Furthermore"
- Line 141: „improves" → „calculates". Please consider changing all other „improve " in this paragraph and the next one to „calculate" or „produce"
- Line 206: „at 3 km", 3km resolution? Or 3km height? Please specify it.
- Line 208: The sentence „Increment in wind and hydrometeors show similar patterns, depending on the DA method". Please consider rewriting this and the similar sentences in this section. The sentence is unclear and ambiguous.
- Line 235: „„the LETKF underestimates the temperature," → „„.there is an underestimation in the temperature derived from the LETKF method,"
- Line 253: „Note the amount of change in the snow mixing ratio", what is the point of this sentence?
- Line 278: „The observed GRS radar CFAD" → „The observed CFAD of the GRS radar"
- Line 294: „In Case 1, The" → „In case 1, the"
- Line 305: „hour prediction" → „forecast hours" please also consider replacing the „prediction" with „forecast" in this and the next section.

---

## Author Comment (AC2)

We thank the reviewer's detailed suggestions and constructive remarks. The authors addressed all of the points made by the reviewer and included a point-by-point response to the reviewer's comments.

**Correspondence to reviewer #1:**

**Specific comments:**

1. The authors compared two very different radar DA methods. Therefore, there might be opposite results attained when tuning the parameters, e.g., number of ensemble members. It might be not suitable to tell which method is better. Especially, it seems that the authors used different radar reflectivity operators in the two methods. Please justify it. How about the results if the authors use the same observation operators for radar radial velocity and reflectivity in the two DA methods? How about the results if LETKF also assimilates water vapor indirectly not through covariance based on ensemble members?

⇒ The authors agree with the reviewer that there maybe different results when combining multitude of options used in this study. However, the purpose of this study is to compare two different radar data assimilation (DA) methods in assimilating winter precipitation over Pyeong-Chang area where intensive observation was performed in 2018. There have been numerous radar DA studies using LETKF or 3DVAR in studying summer precipitation but winter precipitation comparison studies have been seldom performed. The authors argue that 3DVAR and LETKF DA methods comparison is quite unique for winter cases. Large scale, broadly spread stratiform systems are seldom studied but are extensively examined in our research. The difference in radar operator between two DA methods arise from the way DA is performed.

When performing radar reflectivity DA using 3DVAR, direct DA using an adjoint model or indirect DA method of classifying hydrometeors using the model's temperature field can be used (Xiao et al., 2007 ; Wang et al., 2013). However, since the currently developed adjoint model does not fully implement the cold-rain process, the error of the calculated hydrometeor mixing ratios during DA can be large. Therefore, indirect method is commonly applied as was the case in this study. In addition, in variational DA method, only the variables included in the observation operator changes, so even if hydrometeor mixing ratios are calculated through radar reflectivity DA, the water vapor mixing ratio and winds do not change unless prediction is performed. In radar DA, the wind is modified because the radial velocity is assimilated, but the water vapor mixing ratio is not updated, so the dynamic and thermodynamic balance is broken. Therefore, when assimilating 3DVAR radar data, the water vapor mixing ratio was calculated by applying the water vapor mixing ratio observation operator. However, when using the water vapor mixing ratio DA, it should be recognized that the S-band radar is an observation device that cannot observe water vapor, and the equation used for the water vapor mixing ratio DA is an empirically calculated equation.

Since ensemble members are used in LETKF radar DA, the correlation between each variable can be considered. Therefore, reflectivity can be directly assimilated. Although, the water vapor mixing ratio is not included in the observation operator, through cross correlations the water vapor mixing ratio is also calculated. Accordingly, it is common not to consider the water vapor mixing ratio observation operator in most LETKF DA studies. In general however, the area where high reflectivity (about 30 dBZ) is observed can be considered saturated. But when considering only the

correlation between hydrometeor mixing ratios and water vapor, the water vapor mixing ratio may decrease even in the area where high reflectivity is observed. In such cases, even if the mixing ratio of hydrometeors increase through RDA, the surrounding area may be dry and the hydrometeors will not grow and evaporate during the forecast period. Therefore, in order to solve this problem, when radar DA is performed, an observation operator including information on the water vapor mixing ratio must be used.

2. If the reviewer understood correctly, only one forecast was produced in each snowfall case in each DA experiment. The reviewer would suggest that one forecast can be produced in each DA cycle, and then there are enough samples for the authors to conduct statistical evaluation (RMSE, FSS, etc.), which will make a solid study.

⇒ Multiple forecast can be produced if we launch the forecast at each DA cycle, but this can create spin-up issues and adjustment problems. Thus, we think that using the analysis at the first several cycling to conduct the forecast might not be appropriate. We assimilate the radar information in the model, and hope the impact could last for a long time. The large scale forcing weather system over south Korea area needs time to develop during DA cycles, especially for stratiform precipitation. Based on the study of You et al. (2020), 3-hr assimilating period can obtain optimal analysis. Therefore, we did not launch the deterministic forecast in each DA cycle.

You, C.-R.; Chung, K.-S.; Tsai, C.-C. Evaluating the Performance of a Convection-Permitting Model by Using Dual-Polarimetric Radar Parameters: Case Study of SoWMEX IOP8. Remote Sens. 2020, 12, 3004. https://doi.org/10.3390/rs12183004.

3. Sections 3.2, 3.3, and 3.5: Please conduct quantitative analysis and comparison, not just full of "large" "more" "similar" "underestimate" "increase" "relatively small/dry" "low" …

⇒ This has been updated in quantitative terms.

*L234–L235:*

*There is an underestimation in the temperature and dew point in the region **below 500 hPa by up to 2 K and 6 K respectively** derived from the LETKF.*

*L235–L237:*

*This is because reflectivity of more than 30dBZ is observed in this area and the water vapor amount **of about 0.3 g kg−1** and temperature **of about 1.5 K** increase in 3DVAR, but decrease according to the covariance between variables in LETKF (Fig. 6).*

*L238–L40:*

*In Case 2, at altitudes below 700 hPa at all two sites, the LETKF underestimates the temperature of about 2 K, The atmospheric humidity of LETKF is similar to that of observed from 700 to 800 hPa, **but because the temperature and the dew point temperature is 2K lower than observed**, mixing ratio of water vapor is relatively small.*

4. Lines 303-314: What spatial scale did the authors use to calculate FSS? Please examine the sensitivity of different spatial scales and precipitation thresholds.

⇒ The spatial scale used for FSS is 10 km because this is the representative horizontal resolution of AWS network in South Korea. Although not shown, some sensitivity tests on FSS have been performed. We found that the FSS will increase as distance increased. And the FSS decreased as the validation thresholds increased. While validating the hourly rainfall, we checked the rainfall during the forecast periods and took the average value which is about 0.5 mm. If we set a high threshold, the validation on rainfall night be meaningless while winter cases is tested.

5. Figures 12-15: How about the results of CTRL? Please include them for comparison.

⇒ Figures 12-15 with CTRL results have been redrawn and included in the revised manuscript.

**Technical corrections:**

1. Line 120: "component" -> "components"

⇒ The sentence has been modified as follows.

*L118–L 120:*

*B is calculated using the National Meteorological Center (NMC) method (Parrish and Derber, 1992), considering 12-h and 24-h forecasts of winter periods (Dec 01, 2017–Mar 31, 2018) with five members of control variables (u, v wind **components,** surface pressure, temperature, and pseudo-relative humidity).*

2. Line 125: "BE" -> "background error"

⇒ In the previous sentence (L80–L 81), we mentioned that BE means background error. Therefore, we did not modify the sentence.

*L80–L 81:*

*The limitations of 3DVAR however is that the model background error (BE) only considers the climatic BE covariance and it was developed for large-scale observational data (Hamil and Snyder, 2002).*

3. Line 126: Please justify the observation errors used in this study.

⇒ We have corrected the sentence in L126–L 127:

*L126–L 127:*

*The observation errors **including instrumental errors and random errors** for radial wind and reflectivity are assumed as 3 m s−1 and 5 dBZ. Section 2.2.3 describes the observation operator method.*

4. Table 1: Please provide a brief description of prognostic variables.

⇒ The LETKF DA system is able to modify the prognostic variables by background error covariance. And these prognostic variables are listed below: 3-dimensional wind, hydrometer (cloud water, rain, cloud ice, snow and graupel/hail) mixing ratios and number concentration, water vapor, pressure and temperature.

*L126–L 127:*

*For analysis, the system will update the prognostic variables, such as 3D wind, hydrometeor mixing ratio, potential temperature, and geopotential height, using their BE covariances. Besides, the variable-depend horizontal and vertical localization radii are set (Table 1)*

5. Line 188: "compare" -> "examine"

   ⇒ We have corrected the sentence in L188.

   *L 189:*

   *We considered several verification parameters to objectively examine  the model's prediction improvement.*

6. Lines 194-195: Did the authors interpolate station data to the model grid?

   ⇒ While interpolating the model grids to AWS station location, the nearest model grid point is found first. Next, letting the first point as a pivot point, find another three points and make a square, which enclose the AWS point. Lastly, using inverse-distance method and giving the four points different weight using cubic spline to interpolate the values and calculated the statistics.

7. Lines 195 and 201: "using" -> "by"

   ⇒ The sentence has been modified as follows.

   *L 195-196:*

   *where N is the number of horizontal grid points, $P_i$ is the precipitation (mm) in the model's predicted field, and $O_i$ is the precipitation observed by AWS (mm).*

   *L 202-203:*

   *$\bar{P}$ is the mean precipitation (mm) in the model's predicted field, and $\bar{O}$ is the mean precipitation observed by AWS (mm).*

8. Line 196: "fractions skill score" -> "fractions skill score (FSS)"

   ⇒ We have corrected the sentence in L196

   *L 197:*

   *The fractions skill score (FSS), proposed in Roberts and Lean (2008), illustrates a new method for forecast validation*

9. Line 206: "at 3 km" means "at 3-km height" or "at the 3-km domain"

   ⇒ "at 3 km" means "at 3-km height". The sentence has been modified as follows.

   *L 208-209:*

   *Figure 6 shows the radar reflectivity, wind, snow, water vapor amount, and temperature increments at 3-km height  for 0000 UTC, December 24, 2017.*

10. Lines 218-220: Please conduct a quantitative comparison.

   ⇒ The sentence has been modified as follows.

   *L 220-222:*

   *Where a reflectivity of 15 dBZ or more was observed, the inland water vapor mixing ratio is reduced 0.4 g kg−1, the temperature is reduced by 0.8 K in LETKF and in 3DVAR, it increased by 0.2 g/kg and 0.8 K, respectively.*

---

## Author Comment (AC3)

We thank the reviewer's detailed suggestions and constructive remarks. The authors addressed all of the points made by the reviewer and included a point-by-point response to the reviewer's comments.

**Correspondence to reviewer #2:**

General response:

1. This study compares two radar data assimilation methods including 3DVAR and LETKF for ICE-POP snowfall cases. It was found that water vapor adjustment is important for the radar data assimilation. However, water vapor and temperature adjustment has been developed for years. What is new for this conclusion?

   ⇒ We think that 3DVAR and LETKF DA methods comparison is quite unique for winter precipitation cases. Most of the previous works are focused on severe weather systems such as thunderstorms, typhoons and squall lines. Large scale, broadly spread stratiform systems are seldom studied but are extensively examined in our research. We conducted additional studies to investigate how the two different methods operates under the same WRF double moment 6-class microphysics scheme (see Appendix A.). The conclusion leads to not just the importance of water vapor but how they interact with different microphysical terms in winter precipitation systems.

2. It is weird to me that the assimilation of radar observation results constant negative increment of snow mixing ratio. Detail descriptions of reflectivity observation operators are needed. A possible explanation is the current radar data assimilation does not adjust the hydrometer variables. In that case, what is the meaning of reflectivity assimilation?

   ⇒ This results from the fact that CTRL run overestimated the amount of snow heavily in the mid-to-upper levels resulting in stronger reflectivity compared to observations. As a result, the DA systems are trying to reduce the amount of snow mixing ratio. Thus, the current radar DA system do adjust the hydrometeor observations. The difference in radar operator between two DA methods arise from the way DA is performed.

   When performing radar reflectivity data assimilation (DA) using 3DVAR, direct DA using an adjoint model or indirect DA method of classifying hydrometeors using the model's temperature field can be used (Xiao et al., 2007 ; Wang et al., 2013). However, since the currently developed adjoint model does not fully implement the cold-rain process, the error of the calculated hydrometeor mixing ratios during DA can be large. Therefore, indirect method is commonly applied as was the case in this study. In addition, in variational DA method, only the variables included in the observation operator changes, so even if hydrometeor mixing ratios are calculated through radar reflectivity DA, the water vapor mixing ratio and winds do not change unless prediction is performed. In RDA, the wind is modified because the radial velocity is assimilated, but the water vapor mixing ratio is not updated, so the dynamic and thermodynamic balance is broken. Therefore, when assimilating 3DVAR radar data, the water vapor mixing ratio was calculated by applying the water vapor mixing ratio observation operator. However, when using the water vapor mixing ratio DA, it should be recognized that the S-band radar is an observation device that cannot observe water vapor, and the equation used for the water vapor mixing ratio DA is an empirically calculated equation.

   Since ensemble members are used in LETKF radar DA, the correlation between each variable can be considered. Therefore, reflectivity can be directly assimilated. Although, the water vapor mixing

ratio is not included in the observation operator, through cross correlations the water vapor mixing ratio is also calculated. Accordingly, it is common not to consider the water vapor mixing ratio observation operator in most LETKF DA studies. In general however, the area where high reflectivity (about 30 dBZ) is observed can be considered saturated. But when considering only the correlation between hydrometeor mixing ratios and water vapor, the water vapor mixing ratio may decrease even in the area where high reflectivity is observed. In such cases, even if the mixing ratio of hydrometeors increase through RDA, the surrounding area may be dry and the hydrometeors will not grow and evaporate during the forecast period. Therefore, in order to solve this problem, when radar DA is performed, an observation operator including information on the water vapor mixing ratio must be used.

**Minor issues:**

1. Line 15-20: "… a lack of a water vapor and temperature observation operator." What does this mean? I believe the radar data assimilation also adjusts water vapor and temperature in LETKF. The cross variable correlation is an advantage for the common used EnKF. On the contrary, the empirical saturation adjustment of 3DVRA will lead to serious overestimation of precipitation. Please see more in Carlin et al (2017).

   ⇒ The authors agree with the reviewer that LETKF can adjust water vapor and temperature with cross correlations and 3DVAR needs empirical adjustment. However, In LETKF DA, the temperature and the water vapor are adjusted during DA cycling, but the increment is too small, and the authors argue that it is not enough. The detail is described in the response to general comments #2.

2. For section 2.2.3, how is the reflectivity data assimilated? Any adjustment of hydrometer variables for 3DVAR and LETKF? Detail descriptions are needed. Most important, for snowfall radar data assimilation cases, I think the direct adjustment of snow mixing ration for radar data assimilation are necessary. Otherwise, any explanations of the improvement of snow are hardly convincible.

   ⇒ The research purpose of this study was to compare two different radar DA (RDA) methods in assimilating winter precipitation over Pyeong-Chang area where intensive observation was performed in 2018. This paper addressed the nature of the differences in radar operator between two DA methods that arise from the way DA is performed. More detail of the radar operators for LETKF and 3DVAR can be found in the Appendix B.

3. About the LETKF, what are the analysis variables?

   ⇒ The LETKF DA system is able to modify the prognostic variable by background error covariance. And these prognostic variables are : 3-dimensional wind, hydrometers (cloud water, rain, cloud ice, snow and graupel/hail) mixing ratio and number concentration, water vapor, pressure and temperature.

   *L126–L 127:*

*For analysis, the system will update the prognostic variables, such as 3D wind, hydrometeor mixing ratio, potential temperature, and geopotential height, using their BE covariances. Besides, the variable-depend horizontal and vertical localization radii are set (Table 1)*

4. Line 172: 30 members? But it shows 50 members in Figure 4. How the initial members generated?

⇒ The ensemble members used were 50. The sentence has been modified as follows. The ensemble members are perturbed by the CV3 method, which is done by WRF4v.1.3. The cv3 will perturb some variables such as, stream function, unbalanced velocity potential function, water vapor mixing ratio and unbalanced surface pressure. And then 50 ensemble members are formed.
*L 171-172:*

*In the 3DVAR experiment, RDA is performed by setting CTRL as the initial field at the start of the DA period, but LETKF generates 50 ensembles the first time and predicts each ensemble until the first DA (Table 2).*

5. The spin-up period before the radar data assimilation?

⇒ For Case 1, 50 ensemble members start the forecast at 0000UTC 23rd December, 2017 and spin-up for 21 hours. And then conduct the DA at 2100UTC. For case2, the ensemble begin the forecast at 0000UTC 18th March, 2018 and spin-up for 9hours. And then conduct the DA at 0900UTC.

6. Line 195-200: A distance? What value?

⇒ While calculating the FSS, a distance need to be set for the AWS stations validation. Contrast to ETS scores, which is point to point validation, the FSS consider a tolerance distance for model. As we also know that the model has the spatial and time lag comparing to the true observation. Thus, while interpolating the model grids to AWS station location, the nearest model grid point is found first. Next, letting the first point as a pivot point, find another three points and make a square, which enclose the AWS point. Lastly, using inverse-distance method and giving the four points different weight using cubic spline to interpolate the values and calculated the statistics.

7. Line 208-209, "…, depending on the DA method". What does this mean?

⇒ The sentence has been modified accordingly. It means 3DVAR and LETKF are showing similar improvements on snow mixing ratio and wind.
*L 210-211:*

*Regardless of the DA method, increments in wind and hydrometeors show similar patterns.*

8. For Figure 6, the final analysis or the first analysis? Any explanation why both LETKF and 3DVAR get negative increment of snow mixing ratio?

⇒ Please refer to general comment response #2.

9. For Figure 9, in what region? Key information should be clearly stated in the figure caption.

⇒ We have corrected the sentence in L196

*L 254-257:*

*Figure 9 shows the difference in the amount of water vapor and hydrometeor mixing ratio vertical profile in the analysis field between experiments. (a), (d) is the difference between LETKF and CTRL, (b), (e) between 3DVAR and CTRL, and (c), (f) between LETKF-3DVAR **in red area.***

*Figure 9 caption:*

*Figure 9: Vertical profiles of water vapour(black line) and five hydrometeor mixing ratios(cloud: red, rain: yellow, ice: green, snow: blue and graupel: purple line) differences between (a), (d) LETKF and CTRL, (b), (e) 3DVAR and CTRL, (c), (f) LETKF and 3DVAR **in red area** for (a)–(c) 0000 UTC, December 24, 2017, (d)–(f) 1200 UTC, March 18, 2018, respectively.*

10. Please rewrite section 3. Studies have already compared the difference between 3DVAR and LETKF. What is new for this comparison?

⇒ Please refer to general comment response #1.

---

## Author Comment (AC4)

We thank the reviewer's detailed suggestions and constructive remarks. The authors addressed all of the points made by the reviewer and included a point-by-point response to the reviewer's comments.

**Correspondence to reviewer #3:**

**General comments:**

The paper uses radar data to run two different data assimilation methods, LETKF and 3dVAR, during two snowfall events observed in ICE-POP 2018 fleld campaign. The authors compared the analysis and the forecasted data of different variables including wind, water vapor, temperature, and snowfall to show the importance of water vapor assimilation. The logic and structure of the paper are clear and easy to follow. However, several sentences need to be rewritten/reconsidered as well as some concerns which need to be considered before publication.

**Specific comments:**

1. It seems the authors run only one forecast cycle to compare two methods which cause few samples to do the verification methods. Please consider running more forecast cycles to have enough samples which make the verification as well as the results more reasonable.

   ⇒ Multiple forecast can be produced if we launch the forecast at each DA cycle, but this can create spin-up issues and adjustment problems. Thus, we think that using the analysis at the first several cycling to conduct the forecast might not be appropriate. We assimilate the radar information in the model, and hope the impact could last for a long time. The large scale forcing weather system over south Korea area needs time to develop during DA cycles, especially for stratiform precipitation. Based on the study of You et al. (2020), 3-hr assimilating period can obtain optimal analysis. Therefore, we did not launch the deterministic forecast in each DA cycle.

   You, C.-R.; Chung, K.-S.; Tsai, C.-C. Evaluating the Performance of a Convection-Permitting Model by Using Dual-Polarimetric Radar Parameters: Case Study of SoWMEX IOP8. Remote Sens. 2020, 12, 3004. https://doi.org/10.3390/rs12183004

2. The authors use two different data assimilation methods; however, in many sentences particularly in the results and summary sections, the "simulation" word was used to refer to the LETKF or 3DVAR methods. Please note that this is the wrong word referring to the assimilation method. The LETKF and 3DVAR do the "assimilation" not "simulation". Sentences like „The snowfall in GWD was less simulated in LETKF" are logically wrong and need to be rewritten.

   ⇒ Thank you for the comment. Because we used the experiment naming the same as the DA method, we noticed it can be confusing. For simulations we will specify that it is from an experiment model run.

3. The word "underestimate" was used many times in sections 3 and 4 to compare two assimilation methods. Since none of these methods were considered as a reference experiment or reference data, using the words "underestimation"or "overestimation" for this comparison is meaningless. The words "underestimation" or "overestimation" could be used when the results are compared with reference data such as observation. Please consider rewriting these sentences.

⇒ The sentence has been modified as follows.

*L 233-234:*

*There is an underestimation in the temperature and dew point in the region below 500 hPa by up to 2 K and 6 K respectively derived from the LETKF*

*L 238-242:*

*In Case 2, at altitudes below 700 hPa at all two sites, the LETKF underestimates the temperature of about 2 K, The atmospheric humidity of LETKF is similar to that of observed from 700 to 800 hPa, but because the temperature and the dew point temperature is 2K lower than observed, mixing ratio of water vapor is relatively small. A relatively dry area exist from ground to 900 hPa, and this area is also simulated more accurately by 3DVAR than LETKF.*

4. In section 2.2.2 the radial wind and the reflectivity errors are assumed 3 ms-1 and 5 dbz respectively; however, the authors did not mention the source of these numbers.

⇒ To conduct the data assimilation(DA), observation errors are needed to be set. Generally, the observation error contains the representative error, instrumental error, and data processing and quality control error. The radar system has it owns instrumental error. And there can be some errors included while the data are being quality controlled and/or interpolated to coarser superobing or data thinning. As a result, considerable values of observation error are set for reflectivity and radial winds. We added the following reference of You et al. 2019 and Do et al. 2022.

> You, C.-R.; Chung, K.-S.; Tsai, C.-C. Evaluating the Performance of a Convection-Permitting Model by Using Dual-Polarimetric Radar Parameters: Case Study of SoWMEX IOP8. Remote Sens. 2020, 12, 3004. https://doi.org/10.3390/rs12183004
>
> Do, P.-N., K.-S. Chung, P.-L. Lin, C.-Y. Ke, and S. M. Ellis, 2022: Assimilating Retrieved Water Vapor and Radar Data from NCAR S-PolKa: Performance and Validation Using Real Cases. Mon. Weather Rev., 1177–1199.

*L 126-127:*

*The observation errors including instrumental errors and random errors for radial wind and reflectivity are assumed as 3 m s−1 and 5 dBZ (You et al., 2020; Do et al.,2022).*

5. In section 2.3 was mentioned that the precipitation up to 24.8 mm was recorded in the red box area from 00 to 12 UTC. It is an unclear sentence. The precipitation reported from an SYNOP station (which probably is the concern of the author in this sentence) would be for a specific point not for a whole specific area. The sentence could be rewritten by pointing to the minimum and maximum precipitation amount in this area as well as the location of the station which had a maximum report of precipitation. Please consider also the last paragraph in this section which has the same problem.

⇒ The exact station name was specified and latitude and longitude information of the station was added. The sentence has been modified as follows.

*L 178-180:*

*As the center of the cyclone passed through southern Korean Peninsula (Fig. 5 (a)), 13.4-cm snow was recorded in Daegwallyeong, and precipitation up to 24.827.0 mm was recorded in at the*

*Deogyu mountain (in red boxed area: latitude: 35.894, longitude: 127.773) from 0000 UTC to 1200 UTC on December 24, 2017.*

*L 183-186:*

*In total, 8.0-cm snowfall was observed in Yongpyong, and a maximum of 15.5-mm precipitation was recorded at the Sabuk (in the blue boxed area : latitude: 37.220, longitude: 128.821), and a maximum of 64.5-mm precipitation was recorded at the Cho island (in red boxed area: latitude: 34.238, longitude: 127.244) from 1200 UTC on March 18 to 0000 UTC March 19, 2018.*

**Table 2.** Selected cases for numerical experiments and their storm characteristics.

| | | | | Maximum accumulated precipitation | |
|---|---|---|---|---|---|
| **Characteristics** | **Total period** | **Data assimilation (DA) period** | **Forecast period** | **Southern part of the Korean Peninsula (red box)** | **Pyeongchang region (blue box),** |
| Case 1 | Shallow precipitation system, (Warm Low✱) | 0000 UTC, Dec 23, 2017– 1200 UTC, Dec 25, 2017 | 2100 UTC, Dec 23, 2017– 0000 UTC, Dec 24, 2017 | 0000 UTC, Dec 24, 2017– 1200 UTC, Dec 25, 2017 | 27.0 mm (Deogyu mountain: latitude: 35.894 longitude: 127.773) | 16.0 mm (Guryongnyeong: latitude: 37.879 longitude: 128.515) |
| Case 2 | Shallow precipitation system, (Warm Low*) | 0000 UTC, Mar 18, 2018– 0000 UTC, Mar 19, 2018 | 0900–12000 UTC, Mar 18, 2018 | 1200 UTC, Mar 18, 2018– 0000 UTC, Mar 19, 2018 | 64.5 mm (Cho island: latitude: 34.238 longitude: 127.244) | 15.5 mm (Sabuk: latitude: 37.220 longitude: 128.821) |

6. In section 3.1 the sentence „The snow mixing ratio is higher in LETKF" is a general sentence that of course is not generally correct. Please consider mentioning clearly about the specific case (time/location) where the assimilated snow mixing ratio is higher than assimilated snow mixing ratio in 3DVAR.

   ⇒ The exact station name was specified and latitude and longitude information of the station was added. The sentence has been modified as follows.

   *L 219-221:*

   *The increase in the wind and snow mixing ratio is similar, regardless of the DA method, and the snow mixing ratio increased by 0.2 g kg−1 where a reflectivity of 35 dBZ or more was observed in LETKF, but the increase in the southwestern part of the Korean Peninsula was not evident in 3DVAR, the snow mixing ratio in the analysis field at 1200 UTC, March 18 was higher in the LETKF.*

7. In Fig. 8, there is no explanation for the dashed lines.

   ⇒ Information about each line has been added as in the sentence below.

   *Figure 8 caption:*

   *Figure 8: Skew-T Log-P diagrams in (a), (c) OSAN and (b), (d) MOO for radiosonde (black line), LETKF (red line) and 3DVAR (blue line) for (a)–(b) 0000 UTC, December 24, 2017, (c)–(d) 1200*

*UTC, March 18, 2018, respectively and the solid line represents the temperature and the dased line represents the dew point temperature.*

8.  Fig. 14(d), there is a weird feature regarding the FSS score. The FSS scores for both LETKF and 3DVAR methods are very low at the first forecast hours and they increase after about 6 hours. This is not a common behavior in the FSS score of a forecast validation. It would be good if the authors recheck this case or explain a bit about such a weird behavior of the FSS score.

    ⇒ Please check the figures shown below, which depict the horizontal rainfall distribution by hourly basis. Some difference between the interpolated AWS rainfall distribution and DA forecasts are shown, especially in Pyeong-Chang area (blue boxed region). When we focus on the first three hours of the forecast period, the rainfall distribution of DA simulation cannot capture the exact position of the rainfall. After the first several hours forecast, the rainfall distributions are better and performs well.

[Figure]

**Figure 1 Hourly precipitation (mm) distribution of (a)–(c) 1300 UTC, March 18, 2018, (d)–(f) 1400 UTC, , (g)–(i) 1500 UTC and (j)–(l) 1600 UTC for Case 2. (a), (d), (g) and (j) is AWS observations, (b), (e), (h) and (k) is LETKF, and (c), (f), (i) and (l) 3DVAR experiment, respectively.**

**Technical correction:**

    (1.)    Line 35: „cool ocean winds" → „cold ocean winds"

       ⇒ The sentence has been modified as follows.

       *L 35-37:*

       *The area of the East Sea is approximately 106 km2, and the average depth is 1,800 m, providing relatively cold ocean winds to GWD region in summer, and relatively warm ocean winds in winter, serving as heat storage and supplying water vapor to the atmosphere.*

    (2.)    Line 49: „has" → „get"

       ⇒ The sentence has been modified as follows.

       *L 49-50:*

       *The GWD region get a stronger effect from the East Sea and the mountain range because the slope from the top of the Taebaek Mountains to the East Sea is steep.*

    (3.)    Line 69: „only include information → „include only information"

       ⇒ We have corrected the sentence in L69-71

       *L 69-71:*

       *Because radar data include only information on hydrometeors in the atmosphere and exclude information on water vapor, several radar data assimilation (RDA) studies do not assimilate the water vapor mixing ratio (Chen et al., 2021; Liu et al., 2019; Tong et al., 2020).*

    (4.)    Line88: „Further" → „Furthermore"

       ⇒ The sentence has been modified as follows.

       *L 88:*

       *Furthermore, sampling error can occur when the ensemble members are small.*

    (5.)    Line 141: „improves" → „calculates". Please consider changing all other „improve " in this paragraph and the next one to „calculate" or „produce"

       ⇒ The sentence has been modified as follows.

       *L 140-142:*

       *Because the radar does not directly observe water vapor, the 3DVAR method calculates the water vapor amount and temperature through assumptions based on the empirical relation between relative humidity and reflectivity (Wang et al., 2013).*

       *L 12-14:*

       *LETKF produced the water vapor amount and temperature using the covariance of the ensemble members, but 3DVAR produced the water vapor mixing ratio and temperature through an operator that assumed the atmosphere was saturated when reflectivity was above a certain threshold*

       *L 115-117:*

       *The observed value y (= H(x)) is derived from the observation operator (H) and the observed input value y_0, and the analysis field x is calculated through DA with the initial background field value x_b.*

       *L 148:*

*RH is relative humidity and T is temperature which are control variables calculated through DA.*

*L 337-340:*

*The LETKF experiment considered BEs of ensemble members, and the 3DVAR experiment considered only climatic BEs. LETKF produced the water vapor amount and temperature using the covariance of the ensemble members, but 3DVAR produced the water vapor amount and temperature through an operator that assumed the atmosphere was saturated when reflectivity was above a certain threshold.*

(6.) Line 206: „at 3 km", 3km resolution? Or 3km height? Please specify it.

⇒ The sentence has been modified as follows.

*L 208-209:*

*Figure 6 shows the radar reflectivity, wind, snow, water vapor amount, and temperature increments at 3-km height for 0000 UTC, December 24, 2017*

(7.) Line 208: The sentence „Increment in wind and hydrometeors show similar patterns, depending on the DA method". Please consider rewriting this and the similar sentences in this section. The sentence is unclear and ambiguous.

⇒ The sentence has been modified accordingly. It means 3DVAR and LETKF are showing similar improvements on snow mixing ratio and wind.

*L 219-221:*

*The increase in the wind and snow mixing ratio is similar, regardless of the DA method, and the snow mixing ratio increased by 0.2 g kg−1 where a reflectivity of 35 dBZ or more was observed in LETKF, but the increase in the southwestern part of the Korean Peninsula was not evident in 3DVAR, the snow mixing ratio in the analysis field at 1200 UTC, March 18 was higher in the LETKF.*

(8.) Line 235: „„the LETKF underestimates the temperature," → „.there is an underestimation in the temperature derived from the LETKF method,"

⇒ The sentence has been modified as follows.

*L 232-233:*

*There is an underestimation in the temperature and dew point in the region below 500 hPa by up to 2 K and 6 K respectively derived from the LETKF*

(9.) Line 253: „Note the amount of change in the snow mixing ratio", what is the point of this sentence?

⇒ The sentence has been modified as follows.

*L 258-259:*

*In LETKF the maximum difference of the snow mixing ratio come to a decrease of 0.17 g kg−1 comparing to CRTL, however, the 3DVAR, it decreased by 0.22 g kg−1.*

(10.) Line 278: „The observed GRS radar CFAD" → „The observed CFAD of the GRS radar"

⇒ We have corrected the sentence in L 282-284

*L 282-284:*

*The observed CFAD of the GRS radar shows that the reflectivity increases from 10 km for Case 1, indicating that the hydrometeors formed from above 10 km and slowly grew as they reached the ground (Fig. 12(a)).*

(11.) Line 294: „In Case 1, The" → „In case 1, the"

⇒ Changed accordingly.

*L 300-302:*

*In Case 1, the snowfall in GWD was less simulated in LETKF, whereas snowfall of 10.0 mm or more was simulated in 3DVAR, showing a precipitation pattern similar to the observation (Fig. 13(a)-(c)).*

(12.) Line 305: „hour prediction" → „forecast hours" please also consider replacing the „prediction" with „forecast" in this and the next section.

⇒ The sentence has been modified as follows.

*L 310-311:*

*In Case 1, the 3DVAR FSS score showed a high value of 0.6 or more in all and blue regions from 2 to 4 forecast hours.*

*L 311-312:*

*However, the accuracy of 3DVAR decreases after 4 hours, and LETKF showed higher FSS value than 3DVAR in the 8 forecast hours in the all-region and the 7 forecast hours in the blue region.*

*L 317:*

*Even in Case 1, the FSS score in the 1 forecast hours of 3DVAR was as low as 0.49.*

---

## Author Comment (AC5)

**Appendix A.: Microphysical process budgets**

The figure below shows the vertically integrated microphysical process budgets of CTRL, LETKF, and 3DVAR in the blue region (Pyeong-Chang area) during the forecast period. Among the 39 microphysical processes, the top 10 processes are selected from each experimental group, and only the most common processes are shown. The color of the arrow at the top of the figure indicates from which hydrometeor it grew, and the background color indicates what kind of hydrometeor has grown. Red represents water vapor, yellow represents rain, orange represents cloud, green represents ice, blue represents snow, and purple represents graupel. In Case 1, the pidep process, in which water vapor is deposited into ice, was the most dominant in the three experimental groups, and the main precipitation formation process was the process of ice produced through the previous process growing into snow through the psaci and psaut processes and then melting with rain (psmlt) to form precipitation. In Case 2, the temperature was higher than in Case 1, so the process of condensing water vapor into clouds and accretion as rain was added to the main process shown in Case 1, and it appeared as a major precipitation formation process. In both Cases 1 and 2, the amount of pidep in 3DVAR was approximately twice that of pidep in LETKF. In both cases, the snow mixing ratio of LETFK was higher than 3DVAR in the analysis field of the last data assimilation time, but the amount of psmlt converted from snow to rain was the smallest among the three experimental groups, and the amount was 4.53 g/kg and 2.63 g/kg less than 3DVAR. The amount of microphysical processes growing from water vapor to other hydrometeors, including pidep, was the largest in 3DVAR, where the highest water vapor mixing ratio was calculated at the analysis field, and other formation processes also showed the largest value in 3DVAR. That is, the factor that had the greatest influence on the formation of precipitation during the 12-hour forecast period is the water vapor mixing ratio, and the assimilation of the water vapor mixing ratio is important.

[Figure]

[Figure]

**Figure A.1** Common main microphysical processes of (a) CTRL(black bar), (b) LETKF(red bar) and (c)3DVAR for (a) Case 1, (b) Case 2, the forecast period averaged over blue area (red shading: water vapor formation, yellow shading: cloud water formation, green shading: cloud ice formation, blue shading:snow formation; red arrow: from water vapor, yellow arrow: from cloud water, green arrow: from cloud ice, blue arrow: from snow, purple arrow: from graupel).

The figure below are pie charts showing the proportion of microphysical processes in each experiment group. In all of the experiments and cases, the growth process from water vapor to other hydrometeors showed the largest percentage (red shading) and are similar among two DA methods. This is due to the fact that the activation of hydrometeors are dependent on the microphysical scheme employed in the experiment. However, it can be seen that for Case 1, the process of converting snow and hail into rain (blue and purple shading), which are relatively large hydrometeors, accounted for 22% in 3DVAR, but in LETKF, where the amount of water vapor growing into large hydrometeors was small, the process of converting large snow and hail into rain was 12%.

[Figure]

**Figure A.2** Microphysical process pie chart (a) and (d) CTRL and (b) and (e) LETKF and (c) and(f) 3DVAR for (a)–(c) Case 1, and (d)–(f) Case 2, i.e., the forecast period averaged over blue area (red:

from water vapor, yellow: from cloud water, green: from cloud ice, blue: from snow, purple: from graupel).

**Table A.1** List of cloud microphysical processes for calculating mixing ratios WDM6 scheme.

| Abbreviation | Description |
| --- | --- |
| Pcact | Production rate for activation of cloud condensation nuclei |
| Pgmlt | Production rate for melting of graupel to form rain |
| Pidep | Production rate for (+) deposition/(−) sublimation rate of ice |
| Pigen | Production rate for generation (nucleation) of ice from vapor |
| Pracw | Production rate for accretion of cloud water by rain |
| Prevp | Production rate for (+) condensation/(−) evaporation rate of rain |
| Psaci | Production rate for v of cloud ice by snow |
| Psaut | Production rate for autoconversion of cloud ice to form snow |
| Psdep | Production rate for (+) deposition/(−) sublimation rate of snow |
| Psmlt | Production rate for melting of snow to form cloud water |
| Pgmlt | Production rate for melting of graupel to form cloud water |

**Appendix B.: Reflectivity operator**

**A) The reflectivity operator used in the LETKF.**

While the WRF-LETKF performs, the model output data must be converted into the observational variables such as radial wind and reflectivity. And the operator of reflectivity is refer to Jung et al.2008, 2010. T-matrix based and considering the Rayleigh scattering, a power-law scattering amplitude functions are fitted for S-band radar. Beside the operator also consider the effect of tumbling and tilting while the ice-phase meteors fallen. Also, near the melting layer, the mixing-phase meteors are also considered, which could simulated the bright band effect. The equation demonstrated below is the calculation of reflectivity for rain.

$$Z_{h,r} = \frac{4\lambda^4 \alpha_{ra}^2 N_{0r}}{\pi^4 |K_w|^2} \Lambda_r^{-(2\beta_{ra}+1)} \Gamma(2\beta_{ra}+1)(mm^6 m^{-3}) \quad (1)$$

The $\alpha_{ra}$ and $\beta_{ra}$ are the coefficients of the scattering amplitude function. $\lambda$ is the s-band radar lenghth. $K_w$ is for the dieletric variables. $\Lambda_r$ and $N_{0r}$ are derived by the mixing ratio and total number concentration. The reflectivity of snow and graupel could also be calculated by the similar equation above. Finally, sum up the reflectivity contributed by all the meteors, than the total reflectivity could be converted.

**B) The reflectivity operator used in the 3DVAR.**

The radar reflectivity was partitioned into the reflectivity of each hydrometeor type based on the model background temperature by using the hydrometeor classification method and then converted to the hydrometeor mixing ratio.

The observed reflectivity ($Z_o$) was converted from dBZ to $mm^6 \cdot m^{-3}$, which is the unit for input reflectance ($Z_e$) and is expressed as

$$Z_o = 10 \log_{10} Z_e. \quad (2)$$

$Z_e$ can be expressed as

$$Z_e = Z_r + Z_{ds} + Z_{ws} + Z_g, \quad (3)$$

because it is a volume average that is observed by several hydrometeors, such as rain (r), dry snow (ds), wet snow (ws), and graupel (g) [27–29].

For the precipitation echo data assimilation ($Z_o > -15$ dBZ), Wang et al. (2013) classified hydrometeors by using the model's temperature field (T (K)). Rain exists in a grid with a temperature of $T \geq 5$ °C, and a grid temperature of $-5$°C $< T < 5$ °C assumes that rain, wet snow, hail, and dry snow can coexist (Equations (4)–(7)). The $\alpha$ in Equations (5)–(6) represents a value of zero at $-5$ °C with $\alpha = 1$ at 5 °C, and it varies linearly between zero and one with the model temperature (Equation (8)).

$$Z_e = Z_r \ (5 \ °C \leq T).$$

$$Z_e = \alpha \ Z_r + (1 - \alpha)[Z_{ws} + Z_g] \ (0 \ °C < T < 5 \ °C).$$

$$Z_e = \alpha\ Z_r + (1 - \alpha)[Z_{ds} + Z_g]\ (-5\ °C < T \le 0\ °C).$$

$$Z_e = Z_{ds} + Z_g\ (T \le -5\ °C).$$

$$\alpha = \frac{T + 5°C}{10°C}\quad (-5\ °C < T \le 5\ °C).$$

The reflectivity of the hydrometeors was converted into the mixing ratio $(kg\cdot kg^{-1})$ of each hydrometeor by using the equation of the reflectivity–mixing ratio relationship. The hydrometeor mixing ratio was then used as an indirect assimilation method to assimilate reflectivity into the model [30].

$$q_r = [Z_r\,(\rho_a \times (3.63 \times 10^9)^{-1}]^{0.57}, \tag{9}$$

$$q_{ws} = [Z_{ws}\,(\rho_a \times 10^{11})^{-1}]^{0.57}, \tag{10}$$

$$q_{ds} = [Z_{ds}\,(\rho_a \times (9.80 \times 10^8)^{-1}]^{0.57}, \tag{11}$$

$$q_g = [Z_g\,(\rho_a \times (4.33 \times 10^8)^{-1}]^{0.57}, \tag{12}$$

where $\rho_a$ is the density $(kg\cdot m^{-3})$ of air. To create an environment in which convective clouds are actively maintained, the water vapor mixing ratio was nudged as the saturated water vapor mixing ratio when the observed reflectivity was greater than 30 dBZ. The saturated water vapor mixing ratio was calculated by using the Clausius–Clapeyron equation (e (hPa)) for water, and the water vapor saturation mixing ratio $(q_s)$ was calculated as follows:

$$e = 6.112 \times \exp\left[\frac{L}{R_v}\left(\frac{1}{273.15} - \frac{1}{T}\right)\right], \tag{13}$$

$$q_s = \frac{\epsilon e}{p - (1 - \epsilon)e}, \tag{14}$$

where L is $2.5 \times 10^6$ J·kg$^{-1}$ by heat of evaporation, $R_v$ represents the gas constant of water vapor (461.51 J·kg$^{-1}$·K$^{-1}$), $\epsilon$ is the ratio of the gas constant of dry air to the gas constant of water vapor, and P (hPa) is the pressure of the model.

---

## Author Comment (AC8)

**Appendix A.: Microphysical process budgets**

The figure below shows the vertically integrated microphysical process budgets of CTRL, LETKF, and 3DVAR in the blue region (Pyeong-Chang area) during the forecast period. Among the 39 microphysical processes, the top 10 processes are selected from each experimental group, and only the most common processes are shown. The color of the arrow at the top of the figure indicates from which hydrometeor it grew, and the background color indicates what kind of hydrometeor has grown. Red represents water vapor, yellow represents rain, orange represents cloud, green represents ice, blue represents snow, and purple represents graupel. In Case 1, the pidep process, in which water vapor is deposited into ice, was the most dominant in the three experimental groups, and the main precipitation formation process was the process of ice produced through the previous process growing into snow through the psaci and psaut processes and then melting with rain (psmlt) to form precipitation. In Case 2, the temperature was higher than in Case 1, so the process of condensing water vapor into clouds and accretion as rain was added to the main process shown in Case 1, and it appeared as a major precipitation formation process. In both Cases 1 and 2, the amount of pidep in 3DVAR was approximately twice that of pidep in LETKF. In both cases, the snow mixing ratio of LETFK was higher than 3DVAR in the analysis field of the last data assimilation time, but the amount of psmlt converted from snow to rain was the smallest among the three experimental groups, and the amount was 4.53 g/kg and 2.63 g/kg less than 3DVAR. The amount of microphysical processes growing from water vapor to other hydrometeors, including pidep, was the largest in 3DVAR, where the highest water vapor mixing ratio was calculated at the analysis field, and other formation processes also showed the largest value in 3DVAR. That is, the factor that had the greatest influence on the formation of precipitation during the 12-hour forecast period is the water vapor mixing ratio, and the assimilation of the water vapor mixing ratio is important.

**Figure A.1** Common main microphysical processes of (a) CTRL(black bar), (b) LETKF(red bar) and (c)3DVAR for (a) Case 1, (b) Case 2, the forecast period averaged over blue area (red shading: water vapor formation, yellow shading: cloud water formation, green shading: cloud ice formation, blue shading:snow formation; red arrow: from water vapor, yellow arrow: from cloud water, green arrow: from cloud ice, blue arrow: from snow, purple arrow: from graupel).

The figure below are pie charts showing the proportion of microphysical processes in each experiment group. In all of the experiments and cases, the growth process from water vapor to other hydrometeors showed the largest percentage (red shading) and are similar among two DA methods. This is due to the fact that the activation of hydrometeors are dependent on the microphysical scheme employed in the experiment. However, it can be seen that for Case 1, the process of converting snow and hail into rain (blue and purple shading), which are relatively large hydrometeors, accounted for 22% in 3DVAR, but in LETKF, where the amount of water vapor growing into large hydrometeors was small, the process of converting large snow and hail into rain was 12%.

**Figure A.2** Microphysical process pie chart (a) and (d) CTRL and (b) and (e) LETKF and (c) and(f) 3DVAR for (a)–(c) Case 1, and (d)–(f) Case 2, i.e., the forecast period averaged over blue area (red:

from water vapor, yellow: from cloud water, green: from cloud ice, blue: from snow, purple: from graupel).

| Abbreviation | Description                                                       |
|--------------|-------------------------------------------------------------------|
| Pcact        | Production rate for activation of cloud condensation nuclei       |
| Pgmlt        | Production rate for melting of graupel to form rain               |
| Pidep        | Production rate for (+) deposition/(-) sublimation rate of ice    |
| Pigen        | Production rate for generation (nucleation) of ice from vapor     |
| Pracw        | Production rate for accretion of cloud water by rain              |
| Prevp        | Production rate for (+) condensation/(-) evaporation rate of rain |
| Psaci        | Production rate for v of cloud ice by snow                        |
| Psaut        | Production rate for autoconversion of cloud ice to form snow      |
| Psdep        | Production rate for (+) deposition/(-) sublimation rate of snow   |
| Psmlt        | Production rate for melting of snow to form cloud water           |
| Pgmlt        | Production rate for melting of graupel to form cloud water        |

**Table A.1** List of cloud microphysical processes for calculating mixing ratios WDM6 scheme.

**Appendix B.: Reflectivity operator**

**A) The reflectivity operator used in the LETKF.**

While the WRF-LETKF performs, the model output data must be converted into the observational variables such as radial wind and reflectivity. And the operator of reflectivity is refer to Jung et al.2008, 2010. T-matrix based and considering the Rayleigh scattering, a power-law scattering amplitude functions are fitted for S-band radar. Beside the operator also consider the effect of tumbling and tilting while the ice-phase meteors fallen. Also, near the melting layer, the mixing-phase meteors are also considered, which could simulated the bright band effect. The equation demonstrated below is the calculation of reflectivity for rain.

$$Z_{h,r} = \frac{4\lambda^4 \alpha_{ra}^2 N_{0r}}{\pi^4 |K_w|^2} \Lambda_r^{-(2\beta_{ra}+1)} \Gamma(2\beta_{ra}+1) (mm^6 m^{-3}) \quad (1)$$

The  $\alpha_{ra}$  and  $\beta_{ra}$  are the coefficients of the scattering amplitude function.  $\lambda$  is the sband radar lenghth.  $K_w$  is for the dieletric variables.  $\Lambda_r$  and  $N_{0r}$  are derived by the mixing ratio and total number concentration. The reflectivity of snow and graupel could also be calculated by the similar equation above. Finally, sum up the reflectivity contributed by all the meteors, than the total reflectivity could be converted.

**B) The reflectivity operator used in the 3DVAR.**

The radar reflectivity was partitioned into the reflectivity of each hydrometeor type based on the model background temperature by using the hydrometeor classification method and then converted to the hydrometeor mixing ratio.

The observed reflectivity ( $Z_o$ ) was converted from dBZ to  $mm^{6} \cdot m^{-3}$ , which is the unit for input reflectance ( $Z_e$ ) and is expressed as

$$Z_{\rm o} = 10 \, \log_{10} Z_{\rm e}. \tag{2}$$

Ze can be expressed as

$$Z_e = Z_r + Z_{ds} + Z_{ws} + Z_g, \tag{3}$$

because it is a volume average that is observed by several hydrometeors, such as rain (r), dry snow (ds), wet snow (ws), and graupel (g) [27–29].

For the precipitation echo data assimilation ( $Z_o > -15$  dBZ), Wang et al. (2013) classified hydrometeors by using the model's temperature field (T (K)). Rain exists in a grid with a temperature of  $T \ge 5$  °C, and a grid temperature of  $-5^{\circ}C < T < 5$  °C assumes that rain, wet snow, hail, and dry snow can coexist (Equations (4)–(7)). The  $\alpha$  in Equations (5)–(6) represents a value of zero at  $-5^{\circ}C$  with  $\alpha = 1$  at 5 °C, and it varies linearly between zero and one with the model temperature (Equation (8)).

$$Z_e = Z_r (5 \ ^\circ C \leq T).$$

$$Z_e = \alpha Z_r + (1 - \alpha)[Z_{ws} + Z_g] (0 \circ C < T < 5 \circ C).$$

$$Z_e = \alpha Z_r + (1 - \alpha)[Z_{ds} + Z_g] (-5 \circ C < T \le 0 \circ C).$$

$$Z_e = Z_{ds} + Z_g \ (T \leq -5 \ ^\circ C).$$

$$\alpha = \frac{T + 5^{\circ}C}{10^{\circ}C}$$
 (-5 °C < T ≤ 5 °C)

The reflectivity of the hydrometeors was converted into the mixing ratio  $(kg \cdot kg^{-1})$  of each hydrometeor by using the equation of the reflectivity–mixing ratio relationship. The hydrometeor mixing ratio was then used as an indirect assimilation method to assimilate reflectivity into the model [30].

$$q_{\rm r} = [Z_{\rm r}(\rho_{\rm a} \times (3.63 \times 10^9)^{-1}]^{0.57}, \tag{9}$$

$$q_{ws} = [Z_{ws} (\rho_a \times (4.26 \times 10^{11})^{-1}]^{0.57},$$
(10)

$$q_{ds} = [Z_{ds} (\rho_a \times (9.80 \times 10^8)^{-1}]^{0.57},$$
(11)

$$q_{g} = [Z_{g} (\rho_{a} \times (4.33 \times 10^{8})^{-1}]^{0.57}, \qquad (12)$$

where  $\rho_a$  is the density (kg·m-3) of air. To create an environment in which convective clouds are actively maintained, the water vapor mixing ratio was nudged as the saturated water vapor mixing ratio when the observed reflectivity was greater than 30 dBZ. The saturated water vapor mixing ratio was calculated by using the Clausius–Clapeyron equation (e (hPa)) for water, and the water vapor saturation mixing ratio (qs) was calculated as follows:

$$e = 6.112 \times \exp\left[\frac{L}{R_v}\left(\frac{1}{273.15} - \frac{1}{T}\right)\right],$$
 (13)

$$q_s = \frac{\epsilon e}{p - (1 - \epsilon)e},\tag{14}$$

where L is  $2.5 \times 10^6 \text{ J} \cdot \text{kg}^{-1}$  by heat of evaporation,  $R_v$  represents the gas constant of water vapor (461.51 J \cdot \text{kg}^{-1} \cdot \text{K}^{-1}),  $\epsilon$  is the ratio of the gas constant of dry air to the gas constant of water vapor, and P (hPa) is the pressure of the model.